# DualOptim+: Bridging Shared and Decoupled Optimizer States for Better Machine Unlearning in Large Language Models

Xuyang Zhong [1]   Qizhang Li [2]   Yiwen Guo [2]   Chen Liu [1]

## Abstract

We propose **DualOptim+**, a novel optimization framework for improving machine unlearning in large language models. It introduces a base state to capture common representations shared by forgetting and retaining objectives and delta states to preserve objective-specific residuals. This architecture allows the optimizer to adaptively bridge shared and decoupled states based on the directional conflict between forgetting and retaining gradients. We further introduce DualOptim+ 8bit, a quantized variant that reduces memory overhead without compromising performance. Extensive experiments across fictitious and real-world unlearning, safety alignment, and multi-task learning tasks demonstrate that DualOptim+ consistently achieves a superior trade-off between different objectives. Codes are available at https://github.com/CityU-MLO/DualOptimPlus.

## 1. Introduction

Machine unlearning (MU) (Bourtoule et al., 2021) aims to erase the influence of specific training data, known as the forget set, from pretrained models while preserving their general utility on the retain set. While MU has been extensively studied in computer vision tasks, such as image classification and generation (Heng & Soh, 2023; Kurmanji et al., 2023; Fan et al., 2024b; Huang et al., 2024), the rapid rise of Large Language Models (LLMs) has underscored the need for efficient methods to remove outdated or unauthorized information (Dang, 2021). In this regard, specialized unlearning techniques for LLMs have recently emerged as a significant area of research (Yao et al., 2024; Zhang et al., 2024; Yuan et al., 2025).

Despite recent progress, it is still challenging to balance the erasure of specific information and the preservation of general capability by optimizing various designed unlearning objectives. Most existing LLM unlearning methods (Yao et al., 2024; Zhang et al., 2024; Yuan et al., 2025) minimize the objectives of the forget set and the retain set jointly by the sum of their gradients, which often leads to a significant degradation in model utility. Inspired by Fan et al. (2024b); Huang et al. (2024), alternately optimizing forgetting and retaining objectives shows promising unlearning performance, but it suffers from gradient entanglement for two conflicting objectives when using a shared optimizer state (e.g., the moving average of gradients and squared gradients in Adam). To address this issue, DualOptim (Zhong et al., 2025) decouples optimizer states and uses separate optimizers for different objectives. Despite the effectiveness in computer vision tasks, it yields marginal improvements for LLMs. Therefore, developing a novel updating scheme is essential for effective LLM unlearning.

In this work, we propose **DualOptim+**, a plug-and-play framework compatible with any optimizer with stored states. It introduces a shared **base state** alongside decoupled **delta states** for each optimization objective. Specifically, the base state is updated using gradients from both the forgetting and retaining objectives, enabling it to capture their common representations. In parallel, the delta states are updated by the residual between the objective-specific gradients and the base state, thereby preserving distinct representations unique to each objective. Finally, the parameters are updated by combining the base state and the delta state.

Our theoretical and numerical analyses demonstrate that DualOptim+ functions as an adaptive intermediate between fully shared and decoupled states, adjusting its behavior based on the degree of directional conflict between the forgetting and retaining gradients. We validate the effectiveness of DualOptim+ through extensive experiments across diverse machine unlearning tasks on various LLMs, including fictitious datasets, real-world scenarios, and safety alignment tasks. To mitigate the memory overhead associated with additional optimizer states, we also introduce **DualOptim+ 8bit**. This quantized variant significantly reduces memory consumption, while maintaining the peak

[1]Department of Computer Science, City University of Hong Kong [2]Independent Researcher. Correspondence to: Chen Liu <chen.liu@cityu.edu.hk>, Yiwen Guo <guoyiwen89@gmail.com>.

*Proceedings of the 43rd International Conference on Machine Learning*, Seoul, South Korea. PMLR 306, 2026. Copyright 2026 by the author(s).

performance. Our results indicate that DualOptim+ bridges the gap between decoupled and shared optimizer states to achieve a superior trade-off between forgetting efficacy and model utility. We believe, our method is a generalizable optimization framework, which can be applied in broader scenarios beyond machine unlearning, such as LLM alignment, multi-objective learning, e.t.c.

We summarize the contributions of this paper as follows:

1. We propose **DualOptim+**, which introduces a shared base state to capture common representations and decoupled delta states to preserve task-specific residuals, effectively bridging the gap between shared and decoupled optimizer states.

2. DualOptim+ is a plug-and-play framework applicable to any multi-objective optimization and optimizers with stored states. Theoretical and numerical analysis demonstrate that DualOptim+ functions as an intermediate between fully shared and decoupled states.

3. Extensive experiments across various LLMs, datasets and tasks confirm that our method achieves a superior trade-off between forgetting efficacy and model utility. DualOptim+ 8bit reduces memory overhead by quantization without compromising performance.

## 2. Related Work

Early efforts of machine unlearning (MU) are devoted to tasks such as image classification and image generation (Bourtoule et al., 2021; Heng & Soh, 2023; Jia et al., 2023; Kurmanji et al., 2023; Tarun et al., 2023; Fan et al., 2024b; Huang et al., 2024). It has recently been adapted to address the unique challenges of Large Language Models (LLMs), such as removing sensitive or copyrighted training data (Maini et al., 2024), mitigating harmful behaviors (Li et al., 2024), and efficiently handling the high-dimensional parameter space (Fan et al., 2024a). Based on how the model handles forgotten knowledge, MU methods on LLMs can be categorized into *untargeted* and *targeted* unlearning.

In **untargeted unlearning**, the objective is to eliminate the influence of specific data without constraining the model's subsequent response to the forgotten content. Common techniques for this paradigm include gradient ascent (GA) (Thudi et al., 2021; Yao et al., 2024), negative preference optimization (NPO) (Fan et al., 2024a; Zhang et al., 2024), maximum entropy (ME) (Yuan et al., 2025), and representation misalignment (Li et al., 2024; Zou et al., 2024). Conversely, **targeted unlearning** aims to induce specific model behaviors when encountering forgotten information, such as providing standardized rejection responses (e.g., "I don't know"). It is more user-friendly than untargeted unlearning. Popular methods for targeted unlearning include rejection fine-tuning (IDK) (Maini et al., 2024), direct preference optimization (DPO) (Rafailov et al., 2023), and self-classification (Gandikota et al., 2024).

Besides erasing targeted information, it is also crucial for both untargeted and targeted unlearning to maintain general model utility and avoid catastrophic forgetting. Most existing approaches incorporate cross-entropy loss (Gandikota et al., 2024; Li et al., 2024; Yao et al., 2024; Zhang et al., 2024) or divergence-driven loss (Yuan et al., 2025) to optimize the model utility on a designated retain set.

Most LLM unlearning methods jointly update the objectives for the forget and the retain sets, but this optimization strategy usually leads to excessive forgetting and utility degradation (Zhang et al., 2024). This issue is mitigated by alternatively using gradients from the forgetting and the retaining objectives (Kurmanji et al., 2023; Fan et al., 2024b; Huang et al., 2024). DualOptim (Zhong et al., 2025) further improves the effectiveness and stability of unlearning by using two distinct optimizers with separate states.

In addition to the aforementioned optimization-based methods (Jeung et al., 2025a;b; Reisizadeh et al., 2026), inference-time adjustment methods aim to achieve efficient unlearning without modifying model parameters. Theses methods are mainly based on output intervention (Deng et al., 2025; Wang et al., 2026) and in-context learning (Pawelczyk et al., 2024).

## 3. Method

### 3.1. Preliminaries

Machine unlearning (MU) seeks to solve the following optimization problem:

$$\min_{\boldsymbol{\theta}} \mathcal{L}_f(\boldsymbol{\theta}, \mathcal{D}_f) + \mathcal{L}_r(\boldsymbol{\theta}, \mathcal{D}_r), \tag{1}$$

where $\boldsymbol{\theta}$ represents the model parameter. The forget and retain sets are denoted by $\mathcal{D}_f$ and $\mathcal{D}_r$, respectively. $\mathcal{L}_f$ and $\mathcal{L}_r$ are the corresponding loss functions for forgetting and retaining objectives, respectively. These objectives guide the model to eliminate the information contained in $\mathcal{D}_f$ while simultaneously preserving the utility on $\mathcal{D}_r$.

In the context of large language models (LLMs), most MU methods (Zhang et al., 2024; Yuan et al., 2025) employ the **Joint** updating scheme (see Figure 1 (a)). This scheme sums both the forgetting and retaining objectives and obtains the gradient by a single back-propagation step to minimize (1). The updating step can be formulated as follows. $\mathcal{P}$ represents an optimizer that may store states.

$$\boldsymbol{\theta} \leftarrow \boldsymbol{\theta} - \mathcal{P}(\nabla_{\boldsymbol{\theta}}(\mathcal{L}_f + \mathcal{L}_r)). \tag{2}$$

Some recent works (Fan et al., 2024b; Huang et al., 2024) utilize the **Alternate** updating scheme (see Figure 1 (b))

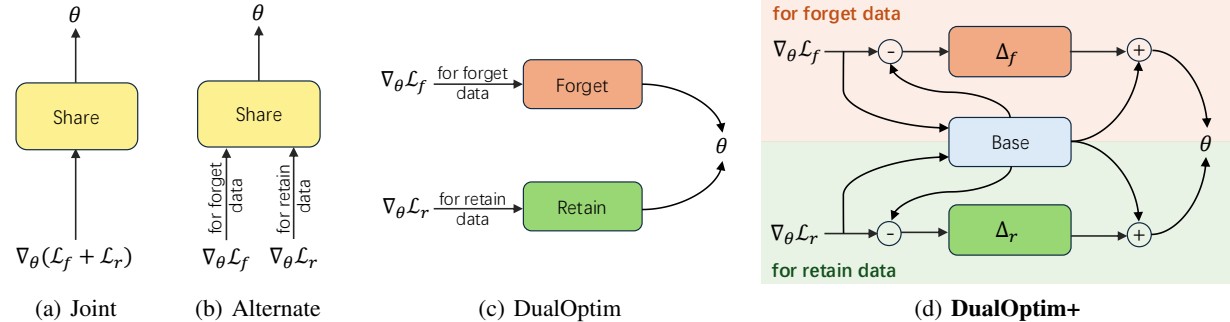

(a) Joint       (b) Alternate       (c) DualOptim       (d) **DualOptim+**

*Figure 1.* **Comparison of baselines and DualOptim+**. The block represents the optimizer state. **(a) Joint** updating scheme sums both the forgetting and retaining losses and executes a single back-propagation step. **(b) Alternate** updating scheme minimizes (1) by gradients from either $\mathcal{L}_f$ or $\mathcal{L}_r$ at each iteration, and alternates between the two objectives. **(c) DualOptim** introduces an independent optimizer state for each objective based on Alternate. **(d) DualOptim+** bridges the shared and decoupled states by introducing base and delta states. The base state is jointly updated by $\nabla_\theta \mathcal{L}_f$ and $\nabla_\theta \mathcal{L}_r$, and $\Delta_f$ / $\Delta_r$ is updated by the difference between $\nabla_\theta \mathcal{L}_f$ / $\nabla_\theta \mathcal{L}_r$ and base state. Ultimately, the momentum term used for the parameter update is reconstructed by combining the base state with the respective delta state.

to improve the performance. This scheme minimizes (1) by gradients from either $\mathcal{L}_f$ or $\mathcal{L}_r$ at each iteration, and alternates between the two objectives. Formally,

$$\begin{cases} \boldsymbol{\theta} \leftarrow \boldsymbol{\theta} - \mathcal{P}(\nabla_{\boldsymbol{\theta}}\mathcal{L}_f) & \text{for forget data} \\ \boldsymbol{\theta} \leftarrow \boldsymbol{\theta} - \mathcal{P}(\nabla_{\boldsymbol{\theta}}\mathcal{L}_r) & \text{for retain data} \end{cases}. \quad (3)$$

Despite its effectiveness, the alternating updating scheme makes MU approaches unstable and sensitive to hyper-parameter tuning. **DualOptim** (Zhong et al., 2025) (see Figure 1 (c)) mitigates the unstable issue and further improves the effectiveness by using two decoupled optimizers $\mathcal{P}_f$, $\mathcal{P}_r$ with separate states. This updating scheme disentangles the conflicting gradients from two objectives. Formally,

$$\begin{cases} \boldsymbol{\theta} \leftarrow \boldsymbol{\theta} - \mathcal{P}_f(\nabla_{\boldsymbol{\theta}}\mathcal{L}_f) & \text{for forget data} \\ \boldsymbol{\theta} \leftarrow \boldsymbol{\theta} - \mathcal{P}_r(\nabla_{\boldsymbol{\theta}}\mathcal{L}_r) & \text{for retain data} \end{cases}. \quad (4)$$

DualOptim is a plug-and-play technique applicable for various unlearning tasks. However, compared with remarkable performance improvement in image classification and generation, applying DualOptim in LLMs yields relatively marginal improvement. Our observations in Figure 2 (a) show that the similarity between the decoupled momentum terms of DualOptim is near zero, implying that the gradients from the forgetting and retaining objectives in LLM unlearning are not always conflicting, especially in the later phase. This phenomenon indicates that we should adaptively utilize both shared features and distinct features from the gradients.

### 3.2. Bridging Shared and Decoupled Optimizer States

In this subsection, we propose **DualOptim+** to bridge the decoupled and shared optimizer states to improve the unlearning performance for LLMs (see Figure 1 (d)). Specifically, DualOptim+ decomposes each optimizer state into two components: a shared **base state** and decoupled **delta**

states for forgetting and retaining objectives. Without loss of generality, we use the first-order momentum term as an example in the analyses below. The updating rule can be straightforwardly extended to other optimizer states.

**Base State.** The base state $B$ is introduced to reserve the common representation shared by the forgetting and retaining objectives. It is updated jointly by $\nabla_\theta \mathcal{L}_f$ and $\nabla_\theta \mathcal{L}_r$, effectively acting as a shared state. Formally,

$$\begin{cases} B \leftarrow \beta B + (1-\beta)\nabla_{\theta}\mathcal{L}_f & \text{for forget data} \\ B \leftarrow \beta B + (1-\beta)\nabla_{\theta}\mathcal{L}_r & \text{for retain data} \end{cases}. \quad (5)$$

where $\beta \in [0, 1)$ is the momentum factor.

**Delta State.** We introduce the delta states $\Delta_f$, $\Delta_r$ to capture the historical residual information: the difference between the individual gradient and the base state. This allows the delta states to reserve the distinct representations specific to the forgetting and retaining objectives. Formally,

$$\begin{cases} \Delta_f \leftarrow \beta\Delta_f + (1-\beta)(\nabla_{\theta}\mathcal{L}_f - \widehat{B}) & \text{for forget data} \\ \Delta_r \leftarrow \beta\Delta_r + (1-\beta)(\nabla_{\theta}\mathcal{L}_r - \widehat{B}) & \text{for retain data} \end{cases}. \quad (6)$$

where $\widehat{B} = B/(1-\beta^t)$ denotes the bias-corrected base state and $t$ is the iteration number.

**Parameter Update.** Finally, the momentum term used for the parameter update is reconstructed by combining the bias-corrected base state $(\widehat{B})$ with the respective bias-corrected delta state $(\widehat{\Delta}_f$ or $\widehat{\Delta}_r)$, i.e., $\widehat{B} + \widehat{\Delta}_f$ or $\widehat{B} + \widehat{\Delta}_r$. For Adam and its variants, the second-order momentum is derived in a similar manner using a separate set of base and delta states updated by the squared gradients $(\nabla_\theta \mathcal{L}_f)^2$ or $(\nabla_\theta \mathcal{L}_r)^2$. The full pseudo-code of DualOptim+ integrated with AdamW is presented in Algorithm 1 where we alternatively utilise $\nabla_\theta \mathcal{L}_f$ for $F_f$ iterations and $\nabla_\theta \mathcal{L}_r$ for $F_r$ iterations. It should be noted that the base state is updated

**Algorithm 1** DualOptim+ with AdamW

---

1: **Input:** parameter $\boldsymbol{\theta}$, learning rate $\eta$, betas $(\beta_1, \beta_2)$, epsilon $\epsilon$, weight decay factor $\lambda$, forget objective $\mathcal{L}_f$, retain objective $\mathcal{L}_r$, total steps $N$, forget frequency $F_f$, retain frequency $F_r$

2: **Initialize:** $\boldsymbol{m}_{\Delta_f} \leftarrow 0, \boldsymbol{m}_{\Delta_r} \leftarrow 0, \boldsymbol{m}_B \leftarrow 0, \boldsymbol{v}_{\Delta_f} \leftarrow 0, \boldsymbol{v}_{\Delta_r} \leftarrow 0, \boldsymbol{v}_B \leftarrow 0, t_f \leftarrow 0, t_r \leftarrow 0$

3: $\widehat{\boldsymbol{m}}_B, \widehat{\boldsymbol{v}}_B \leftarrow \boldsymbol{m}_B, \boldsymbol{v}_B$

4: **for** $t = 1$ **to** $N$ **do**

5:    **if** $t \bmod (F_f + F_r) \leq F_f$ **then**

6:       $t_f \leftarrow t_f + 1$

7:       $\boldsymbol{g}, \boldsymbol{m}_\Delta, \boldsymbol{v}_\Delta, t' \leftarrow \nabla_{\boldsymbol{\theta}}\mathcal{L}_f(\boldsymbol{\theta}), \boldsymbol{m}_{\Delta_f}, \boldsymbol{v}_{\Delta_f}, t_f$

8:    **else**

9:       $t_r \leftarrow t_r + 1$

10:      $\boldsymbol{g}, \boldsymbol{m}_\Delta, \boldsymbol{v}_\Delta, t' \leftarrow \nabla_{\boldsymbol{\theta}}\mathcal{L}_r(\boldsymbol{\theta}), \boldsymbol{m}_{\Delta_r}, \boldsymbol{v}_{\Delta_r}, t_r$

11:    **end if**

12:    $\boldsymbol{\theta} \leftarrow \boldsymbol{\theta} - \eta\lambda\boldsymbol{\theta}$

13:    $\boldsymbol{m}_\Delta \leftarrow \beta_1 \boldsymbol{m}_\Delta + (1 - \beta_1)(\boldsymbol{g} - \widehat{\boldsymbol{m}}_B)$

14:    $\boldsymbol{v}_\Delta \leftarrow \beta_2 \boldsymbol{v}_\Delta + (1 - \beta_2)(\boldsymbol{g}^2 - \widehat{\boldsymbol{v}}_B)$

15:    $\widehat{\boldsymbol{m}}_\Delta, \widehat{\boldsymbol{v}}_\Delta \leftarrow \boldsymbol{m}_\Delta/(1 - \beta_1^{t'}), \boldsymbol{v}_\Delta/(1 - \beta_2^{t'})$

16:    $\boldsymbol{\theta} \leftarrow \boldsymbol{\theta} - \eta(\widehat{\boldsymbol{m}}_B + \widehat{\boldsymbol{m}}_\Delta)/(\sqrt{|\widehat{\boldsymbol{v}}_B + \widehat{\boldsymbol{v}}_\Delta|} + \epsilon)$

17:    $\boldsymbol{m}_B \leftarrow \beta_1 \boldsymbol{m}_B + (1 - \beta_1)\boldsymbol{g}$

18:    $\boldsymbol{v}_B \leftarrow \beta_2 \boldsymbol{v}_B + (1 - \beta_2)\boldsymbol{g}^2$

19:    $\widehat{\boldsymbol{m}}_B, \widehat{\boldsymbol{v}}_B \leftarrow \boldsymbol{m}_B/(1 - \beta_1^t), \boldsymbol{v}_B/(1 - \beta_2^t)$

20: **end for**

21: **Output:** parameter $\boldsymbol{\theta}$

---

after the parameter update to maintain a stable reference for the delta states, thereby enhancing optimization stability.

It should be emphasized that DualOptim+ is a generic framework that can be integrated into any optimizer with stored states. As an additional example, the pseudo-code for DualOptim+ integrated into Muon (Jordan et al., 2024) is provided in Algorithm 2 of Appendix A.

In addition, our method share the similar idea with some federated learning methods (Karimireddy et al., 2020; 2021; Wang et al., 2025; Cheng & Glasgow, 2025), i.e., the base + delta decomposition. However, our method focuses on unlearning task, which is distinct from federated learning in terms of objectives. We defer the detailed discussion and comparison in Appendix D.

### 3.3. Analyses on DualOptim+

The design of DualOptim+ yields a hypothesis regarding its behavior: DualOptim+ acts as **an intermediate between fully shared state in (3) and fully decoupled states in (4)**, adapting based on the degree of directional conflict between

$\nabla_{\boldsymbol{\theta}}\mathcal{L}_f$ and $\nabla_{\boldsymbol{\theta}}\mathcal{L}_r$.

**Theoretical Analysis.** Without the loss of generality, we focus on one coordinate for notation simplicity since DualOptim+ updates the parameters in an elementwise manner. We consider the assumption below before detailed analyses.

**Assumption 3.1. (Gradient Dynamics).** Let $\{g_{f,t} \in \mathbb{R}\}_t$, $\{g_{r,t} \in \mathbb{R}\}_t$ be two sequences representing $\nabla_{\boldsymbol{\theta}}\mathcal{L}_f, \nabla_{\boldsymbol{\theta}}\mathcal{L}_r$ at the time step $t$, respectively. We assume the expectations of $g_{f,t}$ and $g_{r,t}$ over time $\mathbb{E}_t[g_{f,t}] = mG, \mathbb{E}_t[g_{r,t}] = nG$ exist, where $m, n \in [-1, 1]$, and $G$ is a non-negative constant, denoting the largest possible gradient magnitude.

Since the update rules in (5) and (6) involve gradients from two distinct distributions, standard convergence properties for stationary inputs do not directly apply. Nevertheless, the following theorem validates the convergence of both base and delta states in (5) and (6).

**Theorem 3.2.** *(Convergence of Base and Delta States). Considering Assumption 3.1, the update rules (5) and (6), we use $B_t, \Delta_{f,t}, \Delta_{r,t}$ to represent the value of state $B, \Delta_f, \Delta_r$ at the time step $t$, respectively. We have the following asymptotical expectation:*

$$\lim_{T \to \infty} B_{(F_f+F_r)T} = \frac{\beta^{F_r}(1 - \beta^{F_f})m + (1 - \beta^{F_r})n}{1 - \beta^{F_f+F_r}}G,$$

$$\lim_{T \to \infty} \Delta_{f,(F_f+F_r)T} = \frac{F_f\beta^{F_f-1}(1 - \beta)\left(1 - \beta^{F_r}\right)}{\left(1 - \beta^{F_f}\right)\left(1 - \beta^{F_f+F_r}\right)}(m - n)G,$$

$$\lim_{T \to \infty} \Delta_{r,(F_f+F_r)T} = \frac{F_r\beta^{F_r-1}(1 - \beta)\left(1 - \beta^{F_f}\right)}{\left(1 - \beta^{F_r}\right)\left(1 - \beta^{F_f+F_r}\right)}(n - m)G.$$

$$(7)$$

*where $F_f$ and $F_r$, denote the forget frequency and retain frequency respectively in Algorithm 1.*

The proof is deferred to Appendix B.1. According to Theorem 3.2, we can clearly see the expected magnitude of the base state is an interpolation of the expected gradients from the forgetting and the retaining objectives, while the delta states are closely related to their differences. This is consistent with the motivation of DualOptim+. Specifically, we highlight the behavior of DualOptim+ under the boundary conditions determined by $m$ and $n$:

1. **Positive Correlation:** When the expectations of gradients are identical $m = n$, the base state $B$ converges to $mG$, and the delta states $\Delta_f$ and $\Delta_r$ converge to 0. In this case, **DualOptim+ acts like Alternate**.

2. **Negative Correlation:** When the expectations of gradients are negatively proportional with a specific factor $m = -\frac{1-\beta^{F_r}}{\beta^{F_r}(1-\beta^{F_f})}n$, the base state $B$ converges to 0. In this case, **DualOptim+ acts like DualOptim**.

**Numerical Analysis.** We conduct numerical analysis to validate the motivation of DualOptim+. In Figure 2 (a),

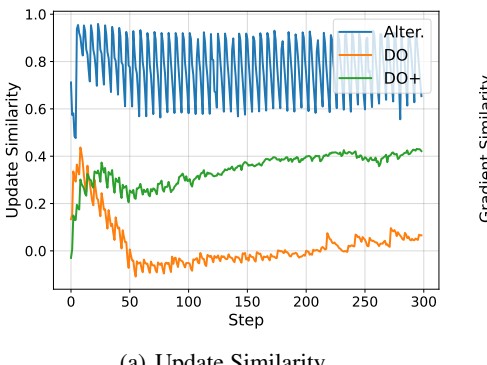

(a) Update Similarity

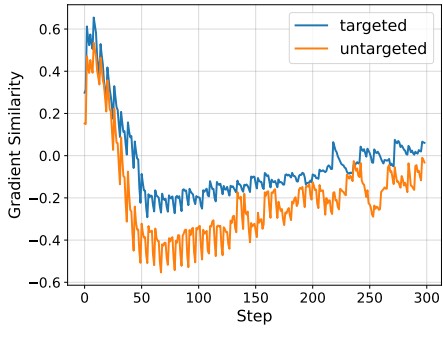

(b) Gradient Similarity

*Figure 2.* Comparison of cosine similarity over time steps. **(a)** Similarity between the update terms for forgetting and retaining of different methods using targeted unlearning objective (Yuan et al., 2025). **(b)** Similarity between forgetting and retaining gradients of targeted and untargeted unlearning objectives. For better visualization, the similarity is calculated based on the exponential moving average (EMA) of the gradients with a factor of 0.9. Note that the results are collected based on the forget10 task of TOFU (Maini et al., 2024). The model is TOFU-finetuned Phi-1.5 (Li et al., 2023), and the optimizer is AdamW.

we show the cosine similarly between the two consecutive updates from the forgetting term and the retaining term for different algorithms. Alternate uses the shared optimizer term, so the similarity is high. By contrast, DualOptim fully decouples these optimizer states and has low similarity. The similarity for DualOptim+ is between Alternate and DualOptim, consistent with our motivation.

Moreover, as illustrated in Figure 2 (b), the cosine similarity between the forgetting and retaining gradients exhibits significant volatility throughout the process of both targeted and untargeted unlearning. This instability suggests that the relationship between the two objectives is highly dynamic, further underscoring the necessity of an adaptive mechanism to transition between shared and decoupled states.

## 4. Experiments

To demonstrate the effectiveness of DualOptim+ in LLM unlearning, we first define the evaluation metrics employed and then present extensive experimental results on unlearning, safety alignment, and multi-task learning tasks. In addition, a comprehensive ablation study is conducted for further analysis. Detailed implementation settings are provided in Appendix E.

### 4.1. Evaluation Metrics

To comprehensively evaluate the performance of the unlearned model, we adopt the evaluation framework established in Yuan et al. (2025), focusing on **Model Utility (MU)** and **Forget Efficacy (FE)**, which are calculated over the retain and forget sets, respectively. These aggregated metrics are based on six primary indicators, including ROUGE (R), Probability (P), Truth Ratio (TR), Token Entropy (TE), Cosine Similarity (CS), and Entailment Score (ES), except that TE is excluded from the calculation of FE.

However, standard FE primarily measures the discrepancy between the model's output and the ground truth in the forget set. While suitable for untargeted unlearning, this is insufficient for targeted unlearning scenarios where the model is expected to provide a specific type of reject response. To address this, we introduce **Targeted Forget Efficacy (TFE)**, which quantifies the similarity between the model's output and predefined rejection templates (e.g., "Sorry, I don't know"). Given that rejection responses are randomly sampled from these templates, providing no fixed ground truth, we define TFE as the harmonic mean of Token Entropy (TE) and Entailment Score (ES) relative to the rejection response. Consequently, we rename the original Forget Efficacy as **Untargeted Forget Efficacy (UFE)**. In addition, we list some examples and their corresponding UFEs and TFEs in Appendix F.

As a result, when evaluating the forget efficacy, we use the average of TFE and UFE for targeted unlearning and merely UFE for untargeted unlearning. We define the **Overall Performance (OVR)** as the average of forget efficacy and model utility, so we have $OVR = 0.25 \times (TFE + UFE) + 0.5 \times MU$ for targeted unlearning and $OVR = 0.5 \times (UFE + MU)$ for untargeted unlearning.

### 4.2. Unlearning in Fictitious Scenario

We start with evaluating our method on the standard TOFU benchmark (Maini et al., 2024). TOFU simulates an ideal scenario with full data access, featuring 200 fictitious authors (20 QA pairs each). It includes three tasks: forget01, forget05, and forget10, targeting the removal of 1%, 5%, and 10% of the data, respectively. The remaining data constitutes the retain set. Additionally, "Real Authors" and "World Facts" sets are used to evaluate general knowledge utility. We employ Phi-1.5-1.3B (Li et al., 2023) and Llama-2-7B (Touvron et al., 2023) released by TOFU as the target

*Table 1.* Performance comparison of different methods on TOFU-finetuned **Phi 1.5** and **Llama 2**. **IDK+GD** (targeted) and **ME+GD** (untargeted) are adopted as the loss functions for unlearning. The results include Untargeted Forget Efficacy (UFE), Targeted Forget Efficacy (TFE), Model Utility (MU) and the Overall Performance (OVR) for forget 1%, 5% data, and 10% data. Note that the reported results are the average of the results obtained from 5 runs with different forget sets.

| | | Phi 1.5 | | | | | | | | | | | |
|---|---|---|---|---|---|---|---|---|---|---|---|---|---|
| | | forget 1% data | | | | forget 5% data | | | | forget 10% data | | | |
| **Loss** | **Method** | UFE ↑ | TFE ↑ | MU ↑ | OVR ↑ | UFE ↑ | TFE ↑ | MU ↑ | OVR ↑ | UFE ↑ | TFE ↑ | MU ↑ | OVR ↑ |
| IDK+GD | Joint | **78.11** | 45.45 | 18.61 | 40.19 | **72.55** | 58.32 | 36.26 | 50.85 | **71.65** | 64.39 | 33.92 | 50.97 |
| IDK+GD | Alternate | 73.35 | 62.49 | 48.14 | 58.03 | 67.73 | 64.30 | 47.81 | 56.91 | 65.82 | 64.46 | 49.54 | 57.34 |
| IDK+GD | DO | 74.75 | 63.51 | 46.46 | 57.80 | 68.49 | 64.34 | 49.50 | 57.96 | 65.87 | **66.84** | 50.25 | 58.30 |
| IDK+GD | DO 8bit | 75.58 | 61.23 | 46.73 | 57.57 | 68.34 | 64.33 | 48.41 | 57.37 | 65.81 | 65.60 | 50.38 | 58.05 |
| IDK+GD | **DO+** | 75.51 | 67.85 | **47.69** | **59.69** | 67.63 | **67.60** | **51.52** | **59.57** | 65.42 | 66.50 | **51.32** | **58.64** |
| IDK+GD | **DO+ 8bit** | 73.69 | 68.36 | 47.53 | 59.28 | 67.56 | 65.94 | 50.36 | 58.55 | 65.26 | 65.59 | 51.30 | 58.36 |
| ME+GD | Joint | **95.41** | – | 11.45 | 53.43 | 91.32 | – | 33.87 | 62.60 | 91.10 | – | 36.88 | 63.99 |
| ME+GD | Alternate | 91.46 | – | 45.78 | 68.62 | 92.30 | – | 49.73 | 71.02 | 91.96 | – | 48.48 | 70.22 |
| ME+GD | DO | 92.79 | – | 45.26 | 69.03 | 91.97 | – | 51.73 | 71.86 | 92.39 | – | 49.23 | 70.81 |
| ME+GD | DO 8bit | 92.83 | – | 45.75 | 69.29 | **93.12** | – | **50.99** | **72.06** | 92.32 | – | 48.04 | 70.19 |
| ME+GD | **DO+** | 93.92 | – | **46.19** | **70.06** | 93.07 | – | 50.87 | 71.97 | 92.46 | – | **50.32** | **71.39** |
| ME+GD | **DO+ 8bit** | 92.48 | – | 46.16 | 69.32 | 93.13 | – | 50.55 | 71.84 | 92.78 | – | 49.96 | 71.37 |

| | | Llama 2 | | | | | | | | | | | |
|---|---|---|---|---|---|---|---|---|---|---|---|---|---|
| | | forget 1% data | | | | forget 5% data | | | | forget 10% data | | | |
| **Loss** | **Method** | UFE ↑ | TFE ↑ | MU ↑ | OVR ↑ | UFE ↑ | TFE ↑ | MU ↑ | OVR ↑ | UFE ↑ | TFE ↑ | MU ↑ | OVR ↑ |
| IDK+GD | Joint | **85.96** | 59.50 | 38.62 | 55.68 | **80.29** | 68.60 | 55.63 | 65.04 | **78.08** | 71.25 | 57.15 | 65.91 |
| IDK+GD | Alternate | 81.97 | 66.41 | 73.93 | 74.06 | 75.79 | 70.27 | 73.93 | 73.48 | 74.16 | 68.15 | 73.98 | 72.57 |
| IDK+GD | DO | 83.12 | 66.41 | 73.83 | 74.30 | 76.58 | 70.27 | 73.64 | 73.53 | 74.62 | 68.15 | 73.15 | 72.27 |
| IDK+GD | DO 8bit | 83.11 | 66.41 | 73.77 | 74.27 | 76.42 | 70.27 | 73.71 | 73.53 | 74.62 | 68.15 | 73.38 | 72.38 |
| IDK+GD | **DO+** | 78.76 | 66.41 | **77.40** | **74.99** | 75.11 | 70.27 | 75.91 | **74.30** | 73.91 | 68.15 | 75.46 | 73.25 |
| IDK+GD | **DO+ 8bit** | 79.05 | 66.41 | 76.53 | 74.63 | 75.04 | 70.27 | 76.27 | **74.42** | 73.49 | 68.15 | **76.14** | **73.48** |
| ME+GD | Joint | 95.89 | – | 59.70 | 77.80 | **97.65** | – | 57.15 | 77.40 | **97.66** | – | 60.63 | 79.15 |
| ME+GD | Alternate | 97.25 | – | 74.89 | 86.07 | 97.07 | – | 75.64 | 86.36 | 96.86 | – | 75.34 | 86.10 |
| ME+GD | DO | 97.46 | – | 75.06 | 86.26 | 96.96 | – | 75.90 | 86.28 | 96.78 | – | 75.60 | 86.19 |
| ME+GD | DO 8bit | 97.76 | – | 75.55 | 86.66 | 96.74 | – | 75.52 | 86.13 | 96.57 | – | 75.44 | 86.01 |
| ME+GD | **DO+** | 97.88 | – | **75.82** | **86.85** | 96.91 | – | 76.09 | **86.50** | 96.85 | – | **75.86** | **86.35** |
| ME+GD | **DO+ 8bit** | 97.50 | – | 75.01 | 86.26 | 96.69 | – | **76.16** | 86.43 | 96.78 | – | 75.52 | 86.15 |

models, which has been fine-tuned on the constructed data to ensure it can exactly gives answers to questions in TOFU.

To evaluate the effectiveness of **DualOptim+ (DO+)**, we compare it against three baselines: **Joint**, **Alternate**, and **DualOptim (DO)** (Zhong et al., 2025). Notably, DO and DO+ require additional memory for optimizer states, specifically $2\times$ and $3\times$ the memory consumption of a single standard Adam, respectively. To mitigate this overhead, we introduce their 8-bit versions, **DO 8bit** and **DO+ 8bit**, implemented using the `bitsandbytes` library to quantize optimizer states. These 8-bit variants significantly reduce memory requirements to $1/2$ and $3/4$ that of a single Adam. To quantify this, we compare the running time and memory consumption of the evaluated methods in Appendix C.1.

The results on IDK+GD (targeted) (Maini et al., 2024) and ME+GD (untargeted) (Yuan et al., 2025) loss functions are reported in Table 1. We can observe that Joint suffers from excessive forgetting, leading to a significant degradation in model utility. In contrast, DualOptim+ (DO+) achieves the best overall performance in most cases, with its superiority

primarily stemming from its ability to preserve model utility while effectively unlearning. Additionally, DO+ yields a larger performance gain over DO in targeted unlearning compared to untargeted unlearning, as the forgetting and retaining objectives are less conflicted in the targeted setting, allowing DO+ to better leverage its optimization advantages as an intermediate state between fully shared state and fully decoupled states. Moreover, as illustrated in Figure 3 of Appendix C.4, DO+ demonstrates consistently better and more stable unlearning performance over time steps compared to the baselines. Regarding efficiency, the proposed DO 8bit and DO+ 8bit variants consume only $1/4$ of the memory required by their standard 32-bit counterparts (DO and DO+) without compromising performance.

Additionally, the results on DPO+GD (Rafailov et al., 2023) and NPO+GD (Zhang et al., 2024) loss functions are presented in Table 12 of Appendix C.2. To further evaluate the efficacy of our method with parameter-efficient fine-tuning techniques, we report the results with LoRA (Hu et al., 2022) in Table 13 of Appendix C.3. These observations are

consistent with that in Table 1, indicating the effectiveness of DualOptim+ in broad scenarios.

## 4.3. Unlearning in Real-world Scenario

We consider a realistic scenario where the unlearning is performed without access to the original training data. Following Liu et al. (2024); Yuan et al. (2025), we identify real-world individuals memorized by Llama-3-8B-Instruct (Grattafiori et al., 2024). We select 20 individuals as unlearning targets, generating the forget set using Llama 3's own responses to 20 questions per person. A neighbor set of 40 additional individuals is selected as the retain set: 20 of which are used for regularization during unlearning, while the remaining 20 are used to evaluate Model Utility. Furthermore, general capability is evaluated via five downstream tasks: ARC-c (Clark et al., 2018), MMLU (Hendrycks et al., 2021), TruthfulQA (Lin et al., 2021), TriviaQA (Joshi et al., 2017), and GSM8K (Cobbe et al., 2021). Joint, Alternate, DualOptim (DO), DualOptim+ (DO+), and the 8bit variants of DO and DO+ are evaluated in the experiment. IDK+GD and ME+GD are adopted as the loss functions for targeted and untargeted unlearning, respectively.

As shown in Table 2, DO+ and its 8-bit variant (DO+ 8bit) consistently achieves effective forgetting while maintaining superior machine utility on the retain set and downstream tasks. Surprisingly, DO underperforms the simpler Alternate approach, particularly in model utility and downstream tasks, suggesting that the fully decoupled framework requires the additional refinements present in DO+ to effectively handle the complexities of real-world data unlearning. Furthermore, compared with untargeted unlearning, the results indicate that targeted unlearning typically leads to a collapse in model utility compared to the initial state, dropping from 61.45 to roughly half that value. This phenomenon is likely to arise from a rigid mapping to specific "I don't know" responses in targeted objective, leading to catastrophic interference with the model's internal representations. In contrast, the untargeted objective merely aims to suppress the ground-truth response through entropy maximization, imposing a "softer" optimization constraint that induces a more marginal representational shift and better preserves the model's underlying reasoning logic.

## 4.4. Safety Alignment

In this subsection, we extend our evaluation to safety alignment (Bai et al., 2022), which can be framed as a specialized unlearning task. The objective is to eliminate unsafe knowledge while preserving model utility. Following the experimental setup of Bianchi et al. (2024), our initial model is Llama-3-8B-Instruct fine-tuned on Alpaca (Taori et al., 2023) to ensure robust instruction-following capabilities. We then simulate the unlearning process by tuning the model on 20,000 Alpaca instructions combined with 2,000 safety instructions; here, the safety and Alpaca instructions serve as the forget and retain sets, respectively. Follow Bianchi et al. (2024), we perform standard SFT on the mixed dataset, which is equivalent to targeted unlearning. The objectives for untargeted unlearning are not considered here, since the objective does not meet the requirements of the task, i.e., reject replying to unsafe queries.

To evaluate the model's harmlessness, we utilize the same collection of harmful instruction datasets as that in Bianchi et al. (2024), including I-MaliciousInstructions (I-Mali) (Taori et al., 2023), I-CoNa (Fanton et al., 2021), I-Controversial (I-Cont) (Bianchi et al., 2024), and Q-Harm (Bai et al., 2022). Response safety is assessed using Llama-Guard-2-8B (Team, 2024), with the final safety rate reported as the primary metric. Model utility is measured following the methodology described in Sec. 4.3. Additionally, we calculate the over-refusal rate on XSTest (Röttger et al., 2024) to identify exaggerated safety behaviors. We compare all methods previously discussed in Sec. 4.2 and 4.3.

As shown in Table 3, DO+ achieves the superior safety-utility trade-off among the evaluated methods. While all unlearning-based approaches significantly improve the model's safety metrics compared with the initial state, DO+ distinguishes itself by attaining the highest overall performance (56.45%) and the highest average utility score (32.81%). This indicates that DO+ is particularly effective at erasing harmful knowledge while preserving the model's core capabilities. Furthermore, DO+ maintains a relatively low over-refusal rate of 28.13% on the XSTest benchmark, which is notably lower than that of the standard DO (30.27%) and Alternate (29.20%) methods. These results suggest that DO+ not only successfully aligns the model with safety requirements but also effectively mitigates the common pitfall of exaggerated safety behaviors.

## 4.5. Multi-task Learning

Our method can be easily extended to multi-task learning tasks. For $N$ tasks, we need to maintain one base state and $N$ delta states. Specifically, we finetune Llama-2-7B on three different tasks, i.e., Py150 (code) (Lu et al., 2021), ScienceQA (science) (Mishra et al., 2022), NumGLUE-cm (math) (Lu et al., 2022). The datasets are collected from TRACE (Wang et al., 2023), and we only evaluate DO 8bit and DO+ 8bit to reduce memory consumption.

The results listed in Table 4 indicate that our method is still effective in the context of multi-task learning. Note that in the context of unlearning, the severity of gradient conflicts gives DO a distinct advantage. Conversely, in multi-task learning, where these conflicts are less pronounced, the gains from DO are negative, whereas DO+ continues to deliver a substantial boost in performance.

*Table 2.* Performance comparison of different methods on **Llama 3** for unlearning real-world data. **IDK+GD** (targeted) and **ME+GD** (untargeted) are adopted as the loss functions for unlearning. The results for unlearning tasks include Untargeted Forget Efficacy (UFE), Targeted Forget Efficacy (TFE), Model Utility (MU) and the Overall Performance (OVR). The average metrics (AVG) on downstream tasks are calculated. Note that the reported results are the average of the results obtained from 3 runs using different random seeds.

| Loss | Method | Unlearning Task | | | | Downstream Tasks | | | | | |
|---|---|---|---|---|---|---|---|---|---|---|---|
| | | UFE ↑ | TFE ↑ | MU ↑ | OVR ↑ | ARC-c ↑ | MMLU ↑ | TruthfulQA ↑ | TriviaQA ↑ | GSM8K ↑ | AVG ↑ |
| | Initial | 30.55 | – | 61.45 | 46.00 | 55.38 | 64.59 | 37.33 | 50.93 | 76.12 | 56.87 |
| IDK+GD | Joint | 85.54 | **72.96** | 27.38 | 53.32 | 46.79 | 62.85 | 33.41 | 7.58 | 74.32 | 44.99 |
| | Alternate | 85.49 | 69.95 | 29.19 | 53.45 | 49.77 | 63.31 | 35.62 | 12.71 | 74.15 | 47.11 |
| | DO | 85.25 | 69.60 | 28.06 | 52.73 | 48.47 | 63.20 | 35.29 | 10.33 | 72.35 | 45.93 |
| | DO 8bit | 85.28 | 69.60 | 27.68 | 52.56 | 48.75 | 63.08 | 35.05 | 11.56 | 72.35 | 46.16 |
| | **DO+** | **85.72** | 69.94 | 27.96 | 52.90 | 50.85 | 64.43 | 36.35 | 11.17 | **76.02** | 47.77 |
| | **DO+ 8bit** | 85.47 | 69.59 | 33.36 | **55.45** | **52.56** | **64.51** | **36.80** | 17.86 | 75.21 | **49.39** |
| ME+GD | Joint | **97.97** | – | 24.53 | 61.25 | 43.29 | 63.46 | 25.05 | 29.61 | 62.34 | 44.75 |
| | Alternate | 97.75 | – | 35.23 | 66.49 | 48.66 | 64.00 | 25.38 | **38.30** | 63.68 | 48.00 |
| | DO | 97.67 | – | 37.51 | 67.60 | 45.36 | 63.27 | 25.34 | 37.45 | 37.07 | 41.69 |
| | DO 8bit | 97.67 | – | 35.42 | 66.55 | 47.95 | 63.67 | 25.95 | 34.20 | 47.58 | 43.87 |
| | **DO+** | 97.85 | – | 48.40 | 73.13 | **56.52** | 64.16 | **34.80** | 28.08 | **73.44** | **51.40** |
| | **DO+ 8bit** | 97.77 | – | **49.29** | **73.52** | 56.45 | 64.14 | 31.29 | 29.22 | 72.38 | 50.70 |

*Table 3.* Performance comparison of different methods on **Alpaca-finetuned Llama 3** for safety alignment. The averages (AVG) of the metrics on safety and utility tasks are calculated, respectively. XSTest is used to evaluate the over-refusal rate. Note that the reported results are the average of the results obtained from 3 runs using different random seeds.

| Method | Safety | | | | | Utility | | | | | | OVR ↑ | XSTest ↓ |
|---|---|---|---|---|---|---|---|---|---|---|---|---|---|
| | I-Mali ↑ | I-CoNa ↑ | I-Cont ↑ | Q-Harm ↑ | AVG ↑ | ARC-c ↑ | MMLU ↑ | TruthfulQA ↑ | TriviaQA ↑ | GSM8K ↑ | AVG ↑ | | |
| Initial | 28.00 | 38.76 | 55.00 | 64.00 | 46.44 | 45.56 | 52.53 | 29.74 | 12.11 | 13.12 | 30.61 | 33.56 | 0.40 |
| Joint | 94.67 | 96.63 | **97.50** | 97.00 | 96.45 | 47.04 | 51.63 | 33.74 | 12.18 | 14.10 | 31.74 | 54.84 | **28.00** |
| Alternate | **97.00** | 97.38 | **97.50** | **99.67** | **97.89** | 46.81 | 50.83 | **34.60** | 13.83 | 12.94 | 31.80 | 55.67 | 29.20 |
| DO | 95.67 | 97.94 | **97.50** | 99.30 | 97.61 | **47.36** | 50.13 | 33.58 | 14.02 | 12.99 | 31.62 | 55.76 | 30.27 |
| DO 8bit | 96.00 | **98.31** | 95.50 | 99.00 | 97.20 | 47.10 | 50.78 | 33.58 | 13.82 | 12.96 | 31.65 | 54.66 | 28.27 |
| **DO+** | 96.00 | 97.56 | **97.50** | 98.67 | 97.43 | 47.27 | **51.89** | 32.25 | **15.39** | 14.23 | **32.81** | **56.45** | 28.13 |
| **DO+ 8bit** | 96.00 | **98.31** | **97.50** | 99.33 | 97.79 | 46.93 | 51.66 | 34.47 | 13.71 | **14.73** | 32.30 | 55.61 | 28.27 |

*Table 4.* Performance in multi-task learning task. We finetune Llama-2-7B on three different tasks, i.e., Py150 (code), ScienceQA (science), NumGLUE-cm (math).

| | Py150 | ScienceQA | NumGLUE-cm | AVG |
|---|---|---|---|---|
| Joint | 61.09 | 92.40 | 41.67 | 65.05 |
| Alternate | 60.74 | 92.05 | 42.86 | 65.22 |
| DO 8bit | 60.36 | 92.25 | 40.48 | 64.36 |
| **DO+ 8bit** | 60.87 | 91.75 | 48.81 | **67.14** |

### 4.6. Ablation Study

In this subsection, we conduct ablation study on DualOptim+ for further analysis. If not specified, all experiments are conducted based on forgetting 5% data of TOFU on Phi1.5 by using IDK+GD loss function. AdamW is adopted as the optimizer.

**Update timing of base state.** In Table 5, we evaluate the impact of the update timing for the base state. Specifically, we compare the performance when the base state is updated at different stages of the optimization step. The results demonstrate that DualOptim+ achieves optimal performance when

the base state is updated after the parameter update. This strategy ensures that the delta states are calculated against a stable and lagged reference, suppressing oscillations during optimization.

*Table 5.* Unlearning performance when the base state is updated at different stages.

| Stage | UFE ↑ | TFE ↑ | MU ↑ | OVR ↑ |
|---|---|---|---|---|
| Before Δ | 67.54 | 66.55 | 50.82 | 58.93 |
| After Δ | **68.24** | 64.38 | 49.97 | 58.14 |
| After θ | 67.63 | **67.60** | **51.52** | **59.57** |

**Updating rule of base state.** In Table 6, we evaluate the performance of DualOptim+ when the base state is updated via different components, specifically comparing raw gradients ($g$) against the difference between gradients and bias-corrected delta states ($g - \widehat{\Delta}$). The results demonstrate that updating the base state directly with gradients yields the best performance, reinforcing its role in capturing the shared representation between tasks.

*Table 6.* Unlearning performance when the base state is updated by different components.

| Updated by | UFE ↑ | TFE ↑ | MU ↑ | OVR ↑ |
|---|---|---|---|---|
| $g$ | 67.63 | **67.60** | **51.52** | **59.57** |
| $g - \widehat{\Delta}$ | **67.78** | 66.74 | 50.89 | 59.08 |

**Momentum factors.** To avoid introducing additional hyper-parameters, we utilize identical momentum factors $(\beta_1, \beta_2)$ for both the base and delta states by default. To evaluate the sensitivity of this configuration, we test two distinct momentum factor sets for AdamW: **Fast** ($\beta_1 = 0.9, \beta_2 = 0.95$, the default) and **Slow** ($\beta_1 = 0.99, \beta_2 = 0.999$). We examine the performance across various combinations of these sets as presented in Table 7. Our results confirm that the default setting, where both states employ the fast momentum factors, achieves the best performance. Notably, applying the slow set to both states leads to a significant performance drop, as the resulting updates are overly conservative for the unlearning task.

*Table 7.* Unlearning performance when adopting different momentum factors. We introduce two sets of momentum factors, Fast ($\beta_1 = 0.9$, $\beta_2 = 0.95$), Slow ($\beta_1 = 0.99$, $\beta_2 = 0.999$). The configuration (F, S) means that the Fast (F) set is used to update the delta states and the Slow (S) set is used to update the base state.

| Momentum factors | UFE ↑ | TFE ↑ | MU ↑ | OVR ↑ |
|---|---|---|---|---|
| (F, F) | 67.63 | **67.60** | **51.52** | **59.57** |
| (F, S) | 67.14 | 67.00 | 50.69 | 58.88 |
| (S, F) | **67.88** | 67.53 | 51.32 | 59.51 |
| (S, S) | 64.90 | 64.27 | 50.60 | 57.59 |

**Quantization.** To decrease memory overhead, we introduce an 8-bit variant of DualOptim+. Table 8 evaluates the impact of quantizing specific optimizer states. The results demonstrate that quantization yields significant memory savings with only acceptable performance degradation. Furthermore, we observe only a marginal performance gap when varying which states are quantized. Therefore, to maximize memory efficiency, we adopt the quantization of all optimizer states as our default configuration.

*Table 8.* Unlearning performance when quantizing different states.

| Quantize | UFE ↑ | TFE ↑ | MU ↑ | OVR ↑ |
|---|---|---|---|---|
| None | 67.63 | 67.60 | 51.52 | 59.57 |
| $B$ | 67.84 | 66.90 | 50.66 | 59.02 |
| $\Delta$ | 67.55 | 66.19 | 50.31 | 58.59 |
| $B + \Delta$ | 67.56 | 65.94 | 50.36 | 58.55 |

**Retain frequency.** In Table 9, we investigate the impact of various retain frequencies ($F_r$) on the Alternate, DualOptim (DO), and DualOptim+ (DO+), while maintaining a fixed forget frequency ($F_f = 1$). The results indicate that the optimal overall performance is achieved at $F_r = 5$ for our

method. While the sensitivity to $F_r$ is relatively low, we observe a general trade-off: a smaller $F_r$ tends to enhance forget efficacy at the expense of model utility, whereas a larger $F_r$ better preserves utility but slightly diminishes unlearning effectiveness. Notably, the improvement across different $F_r$ for our method is stable and is consistently better than Alternate and DO.

*Table 9.* Overall unlearning performance with different retain frequencies $F_r$. The forget frequency is fixed to 1. The total steps of unlearning is 300.

| $F_r$ | 1 | 2 | 4 | 5 | 9 | 14 |
|---|---|---|---|---|---|---|
| Alter. | 55.82 | 56.10 | 56.18 | **56.28** | 55.79 | 55.06 |
| DO | 56.54 | **59.46** | 57.96 | 59.18 | 58.28 | 56.96 |
| **DO+** | 59.13 | 60.05 | 60.19 | **61.30** | 60.85 | 58.16 |

**Hyperparameters of Joint.** In Table 10, we report the results of a grid search over various forgetting loss weights (with a fixed retaining loss weight of 1) and learning rates for the Joint updating scheme. The results indicate that the Joint method achieves its optimal performance using the default hyperparameters. However, even with tuned hyperparameters, the Joint method consistently underperforms compared to other baselines, particularly in terms of maintaining model utility.

*Table 10.* Unlearning performance of Joint with different **(a)** forgetting loss weights, and **(b)** different learning rates.

(a) Forgetting Loss Weight

| Weight | 0.1 | 0.2 | 0.4 | 0.6 | 0.8 | **1.0** |
|---|---|---|---|---|---|---|
| UFE ↑ | 72.89 | 73.41 | 73.59 | 74.59 | 74.08 | **74.78** |
| TFE ↑ | 54.46 | 54.53 | 61.51 | 62.98 | 65.89 | **66.45** |
| MU ↑ | **42.82** | 41.78 | 37.98 | 38.44 | 37.50 | 36.90 |
| OVR ↑ | 53.25 | 52.88 | 52.77 | 53.61 | 53.74 | **53.76** |

(b) Learning Rate

| Weight | 5e-6 | **1e-5** | 2e-5 | 4e-5 | 6e-5 |
|---|---|---|---|---|---|
| UFE ↑ | 72.72 | 74.78 | 75.61 | 80.21 | **81.14** |
| TFE ↑ | 61.62 | 66.45 | **69.37** | 67.94 | 66.55 |
| MU ↑ | 34.79 | **36.90** | 34.94 | 32.95 | 28.10 |
| OVR ↑ | 50.98 | **53.76** | 53.72 | 53.51 | 50.97 |

## 5. Conclusion

In this work, we proposed a novel optimization framework named DualOptim+ for LLM unlearning that utilizes a base state to capture shared representations between forgetting and retaining objectives, alongside delta states that preserve objective-specific residuals. Our extensive evaluation across diverse unlearning scenarios demonstrates that DualOptim+ provides a more stable and effective trade-off between knowledge erasure and utility preservation than existing methods. In future work, we aim to explore the applicability of our method in more general scenarios.

## Impact Statement

Our method is a generalizable optimization framework to help achieve a better trade-off when there are multiple conflicting learning objectives. It can be applied in broad scenarios, including machine unlearning, safety alignment, multitask learning, etc.

## Acknowledgements

This work is supported by City University of Hong Kong (Project No. 9220132, 9229203).

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

## A. Pseudo-code of DualOptim+ with Muon

The pseudo-code of DualOptim+ integrated in Muon (Jordan et al., 2024) is shown in Algorithm 2.

---

**Algorithm 2** DualOptim+ with Muon

---

1: **Input:** parameter $\boldsymbol{\theta}$, learning rate $\eta$, momentum factor $\beta$, forget objective $\mathcal{L}_f$, retain objective $\mathcal{L}_r$, total steps $N$, forget frequency $F_f$, retain frequency $F_r$
2: **Initialize:** $\boldsymbol{m}_{\Delta_f} \leftarrow 0, \boldsymbol{m}_{\Delta_r} \leftarrow 0, \boldsymbol{m}_B \leftarrow 0$
3: **for** $t = 1$ **to** $N$ **do**
4:     **if** $t \bmod (F_f + F_r) \leq F_f$ **then**
5:         $\boldsymbol{g}, \boldsymbol{m}_\Delta \leftarrow \nabla_{\boldsymbol{\theta}}\mathcal{L}_f(\boldsymbol{\theta}), \boldsymbol{m}_{\Delta_f}$
6:     **else**
7:         $\boldsymbol{g}, \boldsymbol{m}_\Delta \leftarrow \nabla_{\boldsymbol{\theta}}\mathcal{L}_r(\boldsymbol{\theta}), \boldsymbol{m}_{\Delta_r}$
8:     **end if**
9:     $\boldsymbol{m}_\Delta \leftarrow \beta\boldsymbol{m}_\Delta + (\boldsymbol{g} - \boldsymbol{m}_B)$
10:     $\boldsymbol{o} \leftarrow \texttt{NewtonSchulz5}(\boldsymbol{m}_B + \boldsymbol{m}_\Delta)$
11:     $\boldsymbol{\theta} = \boldsymbol{\theta} - \eta\boldsymbol{o}$
12:     $\boldsymbol{m}_B \leftarrow \beta\boldsymbol{m}_B + \boldsymbol{g}$
13: **end for**
14: **Output:** parameter $\boldsymbol{\theta}$

---

## B. Proofs

### B.1. Proof of Theorem 3.2

*Proof.* We prove the convergence of $B_t$, $\Delta_f$ and $\Delta_r$ one by one.

**1. Convergence of $B_t$**

Based on the update rule (5), we have the following equation:

$$B_{(F_f+F_r)T} = \beta^{F_f+F_r}B_{(F_f+F_r)(T-1)} + (1-\beta)\left(\sum_{t=1}^{F_f}\beta^{F_f+F_r-t}g_{f,(F_f+F_r)(T-1)+t} + \sum_{t=1}^{F_r}\beta^{F_r-t}g_{r,(F_f+F_r)(T-1)+F_f+t}\right) \tag{8}$$

Based on Assumption 3.1, we can conclude that $\lim_{T\to\infty} B_{(F_f+F_r)T}$ exists, so we let $X_B = \lim_{T\to\infty} B_{(F_f+F_r)T}$. We consider the equation above, take the expectation over $t$, when $T \to \infty$, we have:

$$X_B = \beta^{F_f+F_r}X_B + (1-\beta)\left(\sum_{t=1}^{F_f}\beta^{F_f+F_r-t}\cdot mG + \sum_{t=1}^{F_r}\beta^{F_r-t}\cdot nG\right) \tag{9}$$

Based on the equation above, we have $\lim_{T\to\infty} B_{(F_f+F_r)T} = X_B = \frac{\beta^{F_r}(1-\beta^{F_f})m+(1-\beta^{F_r})n}{1-\beta^{F_f+F_r}}G$.

**2. Convergence of $\Delta_f$**

When $t \in ((F_f + F_r)(T-1) + F_f, (F_f + F_r)T]$, $\Delta_f$ is not updated, so $\Delta_{f,(F_f+F_r)T} = \Delta_{f,(F_f+F_r)(T-1)+F_f}$. Based on the update rule (6), we have the following equation:

$$\Delta_{f,(F_f+F_r)T} = \beta^{F_f}\Delta_{f,(F_f+F_r)(T-1)} + (1-\beta)\cdot\sum_{t=1}^{F_f}\beta^{F_f-t}\left(g_{f,(F_f+F_r)(T-1)+t} - \widehat{B}_{(F_f+F_r)(T-1)+t-1}\right) \tag{10}$$

When $T \to \infty$, and for $1 \leq t \leq F_f$, according to (9), we have:

$$\lim_{T \to \infty} B_{(F_f+F_r)(T-1)+t-1} = \beta^{t-1} X_B + (1-\beta) \sum_{k=1}^{t-1} \beta^{t-1-k} \cdot mG = \beta^{t-1} X_B + (1-\beta^{t-1})mG \qquad (11)$$

When $t \to \infty$, $\widehat{B}_t = B_t/(1-\beta^t) \to B_t$. Let $X_{\Delta_f} = \lim_{T \to \infty} \Delta_{f,(F_f+F_r)T}$, based on (10) and (11), we have:

$$
\begin{aligned}
X_{\Delta_f} &= \beta^{F_f} X_{\Delta_f} + (1-\beta) \cdot \sum_{t=1}^{F_f} \beta^{F_f-t} \left[ mG - \beta^{t-1} X_B - (1-\beta^{t-1})mG \right] \\
&= \beta^{F_f} X_{\Delta_f} + (1-\beta)F_f \beta^{F_f-1} (mG - X_B)
\end{aligned}
\qquad (12)
$$

Based on the equation above, we have $\lim_{T \to \infty} \Delta_{f,(F_f+F_r)T} = X_{\Delta_f} = \frac{F_f \beta^{F_f-1}(1-\beta)(1-\beta^{F_r})(m-n)}{(1-\beta^{F_f})(1-\beta^{F_f+F_r})}G$

### 3. Convergence of $\Delta_r$

Similarly to (10), we have:

$$\Delta_{r,(F_f+F_r)T} = \beta^{F_r} \Delta_{r,(F_f+F_r)(T-1)} + (1-\beta) \cdot \sum_{t=1}^{F_r} \beta^{F_r-t} \left( g_{r,(F_f+F_r)(T-1)+F_f+t} - \widehat{B}_{(F_f+F_r)(T-1)+F_f+t-1} \right) \quad (13)$$

When $T \to \infty$, and for $1 \leq t \leq F_r$, according to (9), we have:

$$
\begin{aligned}
\lim_{T \to \infty} B_{(F_f+F_r)(T-1)+F_f+t-1} &= \beta^{F_f+t-1} X_B + (1-\beta) \left( \sum_{k=1}^{F_f} \beta^{F_f+t-1-k} \cdot mG + \sum_{k=1}^{t-1} \beta^{t-1-k} \cdot nG \right) \\
&= \beta^{F_f+t-1} X_B + \beta^{t-1}(1-\beta^{F_f})mG + (1-\beta^{t-1})nG
\end{aligned}
\qquad (14)
$$

When $t \to \infty$, $\widehat{B}_t = B_t/(1-\beta^t) \to B_t$. Let $X_{\Delta_r} = \lim_{T \to \infty} \Delta_{r,(F_f+F_r)T}$, based on (13) and (14), we have:

$$
\begin{aligned}
X_{\Delta_r} &= \beta^{F_r} X_{\Delta_r} + (1-\beta) \cdot \sum_{t=1}^{F_r} \beta^{F_r-t} \left[ nG - \beta^{F_f+t-1} X_B - \beta^{t-1}(1-\beta^{F_f})mG - (1-\beta^{t-1})nG \right] \\
&= \beta^{F_r} X_{\Delta_r} + (1-\beta)F_r \beta^{F_r-1} \left[ (\beta^{F_f}-1)mG + nG - \beta^{F_f} X_B \right]
\end{aligned}
\qquad (15)
$$

Based on the equation above, we have $\lim_{T \to \infty} \Delta_{r,(F_f+F_r)T} = X_{\Delta_r} = \frac{F_r \beta^{F_r-1}(1-\beta)(1-\beta^{F_f})(n-m)}{(1-\beta^{F_r})(1-\beta^{F_f+F_r})}G$ $\qquad \square$

## C. More Experiment Results

### C.1. Comparison of Running Time and Memory Consumption

*Table 11.* Memory consumption and running time of different methods. The model is Phi1.5-1.3B and the optimizer is AdamW. The total steps of unlearning is 300, the batch size per GPU is 4. All experiments are implemented on two NVIDIA H20 GPUs.

| Method | Joint | Alternate | DO | DO 8bit | **DO+** | **DO+ 8bit** |
|---|---|---|---|---|---|---|
| Memory (GB/GPU) | 33.26 | 33.26 | 36.15 | 33.57 | 39.04 | 33.68 |
| Running Time (s) | 870 | 443 | 439 | 464 | 475 | 468 |

Table 11 provides an overview of the memory consumption and training efficiency for the various methods evaluated. While DO and DO+ introduce additional memory overhead due to their extra optimizer states, peaking at 39.04 GB/GPU for DO+, their 8-bit implementations, DO 8bit and DO+ 8bit, successfully reduce this footprint to 33.57 GB/GPU and 33.68 GB/GPU, respectively. This demonstrates that the 8-bit variants can achieve near-parity in memory usage with the standard

Joint and Alternate baselines, with the slight overhead resulting from maintaining full-precision embedding layers and storing quantization coefficients. Notably, the methods using alternating update scheme offer a nearly $2\times$ speedup over the Joint method. This efficiency gain occurs because the Joint baseline doubles the equivalent batch size by simultaneously processing both forget and retain data in each step.

### C.2. Results on TOFU with DPO+GD and NPO+GD loss functions

As shown in Table 12, DualOptim+ achieves the best overall performance in most cases, consistent with the observations in Table 1. However, compared to IDK+GD and ME+GD, the DPO+GD and NPO+GD configurations fail to achieve either stable forgetting or enhanced model utility. This performance discrepancy can be attributed to the conservative weighting strategy inherent in these methods, which relies heavily on the reference model and limits the optimizer's ability to deviate significantly from the initial state.

*Table 12.* Performance comparison of different methods on TOFU-finetuned **Phi 1.5** and **Llama 2**. **DPO+GD** (targeted) and **NPO+GD** (untargeted) are adopted as the loss functions for unlearning. The results include Untargeted Forget Efficacy (UFE), Targeted Forget Efficacy (TFE), Model Utility (MU) and the Overall Performance (OVR) for forget 1%, 5% data, and 10% data. Note that the reported results are the average of the results obtained from 5 runs with different forget sets.

| | | **Phi 1.5** | | | | | | | | | | | |
| :---: | :---: | :---: | :---: | :---: | :---: | :---: | :---: | :---: | :---: | :---: | :---: | :---: | :---: |
| | | forget 1% data | | | | forget 5% data | | | | forget 10% data | | | |
| **Loss** | **Method** | UFE ↑ | TFE ↑ | MU ↑ | OVR ↑ | UFE ↑ | TFE ↑ | MU ↑ | OVR ↑ | UFE ↑ | TFE ↑ | MU ↑ | OVR ↑ |
| DPO+GD | Joint | 79.43 | 23.74 | 32.08 | 41.83 | **77.25** | 39.32 | 33.54 | 45.91 | **77.67** | 45.46 | 31.48 | 46.52 |
| | Alternate | 78.56 | 35.11 | 48.23 | 52.53 | 74.34 | 47.26 | 49.68 | 55.24 | 74.49 | **57.51** | 49.87 | 57.94 |
| | DO | 81.58 | 55.36 | 46.32 | 57.39 | 74.86 | 47.10 | 50.94 | 55.96 | 72.97 | 52.61 | 50.28 | 56.53 |
| | DO 8bit | **83.77** | 57.56 | 47.73 | 59.20 | 75.20 | 41.12 | 51.12 | 54.64 | 72.76 | 40.95 | 50.02 | 53.44 |
| | **DO+** | 82.92 | **61.90** | 48.05 | **60.23** | 76.04 | **55.56** | 51.33 | **58.57** | 74.67 | 56.46 | **50.78** | **58.18** |
| | **DO+ 8bit** | 83.46 | 57.44 | 47.61 | 59.03 | 75.96 | 50.13 | **51.40** | 57.23 | 73.14 | 46.21 | 50.24 | 54.96 |
| NPO+GD | Joint | **78.87** | – | 29.89 | 54.38 | **74.68** | – | 31.71 | 53.20 | **73.73** | – | 27.78 | 50.76 |
| | Alternate | 74.62 | – | 48.32 | 61.47 | 70.46 | – | 50.17 | 60.32 | 67.81 | – | 49.15 | 58.48 |
| | DO | 74.88 | – | 48.47 | 61.68 | 70.60 | – | 49.89 | 60.25 | 67.33 | – | 48.73 | 58.03 |
| | DO 8bit | 75.51 | – | 48.77 | 62.14 | 70.17 | – | 50.27 | 60.23 | 66.88 | – | 50.07 | 58.48 |
| | **DO+** | 75.80 | – | 48.72 | 62.26 | 71.63 | – | **51.52** | **61.58** | 69.12 | – | 49.71 | **59.42** |
| | **DO+ 8bit** | 76.10 | – | **48.79** | **62.45** | 70.75 | – | 50.63 | 60.69 | 67.11 | – | **51.25** | 59.18 |

| | | **Llama 2** | | | | | | | | | | | |
| :---: | :---: | :---: | :---: | :---: | :---: | :---: | :---: | :---: | :---: | :---: | :---: | :---: | :---: |
| | | forget 1% data | | | | forget 5% data | | | | forget 10% data | | | |
| **Loss** | **Method** | UFE ↑ | TFE ↑ | MU ↑ | OVR ↑ | UFE ↑ | TFE ↑ | MU ↑ | OVR ↑ | UFE ↑ | TFE ↑ | MU ↑ | OVR ↑ |
| DPO+GD | Joint | **92.02** | 45.74 | 44.19 | 56.54 | **88.11** | 45.17 | 66.98 | 66.81 | **85.14** | 49.94 | 65.05 | 66.30 |
| | Alternate | 88.27 | **53.77** | 73.67 | **72.35** | 83.13 | 46.41 | 73.46 | 69.11 | 80.91 | 36.59 | 74.03 | 66.39 |
| | DO | 89.26 | 48.24 | 72.33 | 70.54 | 83.67 | 45.31 | 71.42 | 67.96 | 80.77 | 38.55 | 72.43 | 66.04 |
| | DO 8bit | 89.47 | 46.54 | 72.32 | 70.16 | 84.49 | **48.19** | 73.07 | 69.71 | 82.18 | 40.49 | 73.31 | 67.32 |
| | **DO+** | 79.49 | 45.96 | **77.51** | 70.12 | 77.99 | 41.86 | **76.51** | 68.22 | 78.43 | 47.55 | **77.19** | **70.09** |
| | **DO+ 8bit** | 82.04 | 43.57 | 77.01 | 69.91 | 81.94 | 47.83 | 75.06 | **69.98** | 80.15 | 45.71 | 75.10 | 69.17 |
| NPO+GD | Joint | 77.31 | – | 55.56 | 66.44 | 69.67 | – | 63.12 | 66.40 | 67.90 | – | 64.46 | 66.18 |
| | Alternate | 73.51 | – | 74.40 | 73.96 | 75.32 | – | 75.69 | 75.51 | 72.77 | – | 75.90 | 74.34 |
| | DO | 73.83 | – | 74.51 | 74.17 | 74.31 | – | 74.52 | 74.42 | 71.78 | – | 73.99 | 72.89 |
| | DO 8bit | 73.85 | – | 74.22 | 74.04 | 75.35 | – | 75.11 | 75.24 | 72.42 | – | 75.75 | 74.09 |
| | **DO+** | **78.51** | – | **75.07** | **76.80** | **75.94** | – | 75.37 | 75.65 | 71.95 | – | 73.95 | 72.95 |
| | **DO+ 8bit** | 76.58 | – | 74.91 | 75.75 | 75.42 | – | **76.01** | **75.72** | 73.79 | – | **76.03** | **74.92** |

### C.3. Results on TOFU with LoRA

To evaluate the effectiveness of our method within a parameter-efficient fine-tuning framework, we apply LoRA (Hu et al., 2022) with a rank of 8 to Llama 2. As shown in Table 13, the performance gains from both DualOptim and DualOptim+ are less pronounced than in full-parameter unlearning, but DualOptim+ and its 8bit variant exhibit the best performance in most cases. Furthermore, the results indicate that the performance gap between LoRA and full-parameter tuning widens as the volume of data to be forgotten increases. This is because the limited degrees of freedom in a low-rank subspace are insufficient to encode the complex modifications required for larger-scale unlearning tasks . Conversely, LoRA significantly

mitigates the excessive forgetting on the retain set that is frequently observed when using the Joint updating scheme.

*Table 13.* Performance comparison of different methods on TOFU-finetuned **Llama 2** with **LoRA** (rank = 8). **IDK+GD** (targeted) and **ME+GD** (untargeted) are adopted as the loss functions for unlearning. The results include Untargeted Forget Efficacy (UFE), Targeted Forget Efficacy (TFE), Model Utility (MU) and the Overall Performance (OVR) for forget 1%, 5% data, and 10% data. Note that the reported results are the average of the results obtained from 5 runs with different forget sets.

| | | Llama 2 + LoRA | | | | | | | | | | | |
| :---: | :--- | :---: | :---: | :---: | :---: | :---: | :---: | :---: | :---: | :---: | :---: | :---: | :---: |
| | | forget 1% data | | | | forget 5% data | | | | forget 10% data | | | |
| **Loss** | **Method** | UFE ↑ | TFE ↑ | MU ↑ | OVR ↑ | UFE ↑ | TFE ↑ | MU ↑ | OVR ↑ | UFE ↑ | TFE ↑ | MU ↑ | OVR ↑ |
| IDK+GD | Joint | **73.70** | 59.50 | 73.07 | 69.84 | **70.10** | 68.60 | 68.82 | 69.08 | **64.72** | **67.54** | 64.92 | 65.52 |
| | Alternate | 71.69 | 66.40 | **74.57** | 71.81 | 64.58 | 66.63 | 70.51 | 68.06 | 59.11 | 56.41 | 66.14 | 61.95 |
| | DO | 72.01 | 66.40 | 74.41 | 71.81 | 68.65 | 69.76 | 70.56 | 69.88 | 64.17 | 62.90 | 64.42 | 64.42 |
| | DO 8bit | 72.03 | 66.40 | 74.51 | **71.86** | 68.77 | 69.77 | 70.63 | 69.95 | 64.08 | 63.13 | 64.62 | 64.11 |
| | **DO+** | 72.09 | **66.41** | 73.87 | 71.56 | 68.77 | 70.10 | 73.05 | 71.24 | 61.22 | 62.76 | 69.88 | 65.94 |
| | **DO+ 8bit** | 71.76 | **66.41** | 73.91 | 71.50 | 68.86 | 70.19 | 73.11 | **71.32** | 61.26 | 62.57 | 69.97 | **65.95** |
| ME+GD | Joint | **96.14** | – | 73.92 | 85.03 | 95.01 | – | 75.04 | 85.02 | 93.27 | – | 75.79 | 84.53 |
| | Alternate | 95.78 | – | 75.97 | 85.88 | 92.69 | – | 75.56 | 84.13 | 88.79 | – | 74.62 | 81.71 |
| | DO | 95.87 | – | 76.00 | 85.94 | 94.97 | – | 76.00 | 85.49 | 93.63 | – | 75.57 | 84.60 |
| | DO 8bit | 95.91 | – | 76.09 | 86.00 | **95.11** | – | 76.13 | **85.62** | **93.84** | – | 75.68 | 84.76 |
| | **DO+** | 95.61 | – | **76.46** | **86.04** | 94.89 | – | 76.28 | 85.59 | 93.73 | – | 76.04 | **84.89** |
| | **DO+ 8bit** | 95.04 | – | 76.40 | 85.73 | 94.90 | – | 76.22 | 85.56 | 93.70 | – | **76.05** | 84.88 |

## C.4. Unlearning Metrics and Losses over Time Steps

As depicted in Figure 3 (a) and (b), DualOptim+ exhibits the most effective and stable performance among the evaluated methods. While the Joint updating scheme achieves similar forget efficacy, its Model Utility is substantially lower than other methods. As observed in Figure Figure 3 (c) and (d), both forgetting and retaining losses converge rapidly under the Joint updating scheme, suggesting the model becomes trapped in a suboptimal local minimum. In contrast, methods employing the alternate updating scheme, i.e., Alternate, DualOptim, and DualOptim+, converge more gradually, enhancing model's exploration capability to find superior local minima.

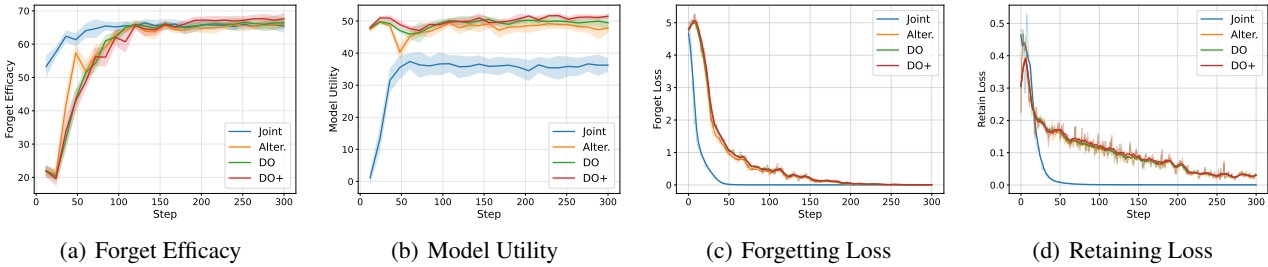

| (a) Forget Efficacy | (b) Model Utility | (c) Forgetting Loss | (d) Retaining Loss |

*Figure 3.* Comparison of unlearning metrics and losses over time steps. We plot the **(a)** forget efficacy, i.e., the average of targeted forget efficacy and untargeted forget efficacy, **(b)** model utility, **(c)** forgetting loss, and **(d)** retaining loss. Note that the results are collected when forgetting 5% data of TOFU using IDK+GD loss function. The model is TOFU-finetuned Phi 1.5.

# D. Comparison with Federated Learning Methods

While federated learning (Konečný et al., 2015) focuses on overcoming client drift caused by non-IID data across a shared objective, machine unlearning must navigate adversarial objectives: the mandate to forget specific information while simultaneously retaining general knowledge. These loss functions are often diametrically opposed, leading to catastrophic forgetting, which is a phenomenon that is not a central concern in standard FL but is the primary bottleneck in unlearning. Our work specifically addresses this tension through a novel optimizer design.

Furthermore, we compare the proposed DO+ with some specific FL methods, i.e., SCAFFOLD (Karimireddy et al., 2020), MIME (Karimireddy et al., 2021), FedCM (Wang et al., 2025), and Local Adam (Cheng & Glasgow, 2025). While FL involves optimizing multiple local models, unlearning tasks require balancing multiple objectives. Therefore, we have to adapt the core design of these FL methods mentioned to the unlearning context to ensure compatibility with our optimization

framework. In the following, we list their implementations and highlight the differences between these adapted methods and DO+ regarding base and delta state updates.

- **SCAFFOLD, MIME:** These two methods share the similar core design. In the context of unlearning tasks, their updating rule of delta state $\Delta$ will be $\Delta \leftarrow \beta\Delta + (1-\beta)(g-B)$, which is the same as DO+. They update the base state $B$ once only after a full forget-retain period (i.e., $F_f + F_r$ batches of data) by the rule $B \leftarrow B + \frac{1}{2}(\widehat{\Delta}_f + \widehat{\Delta}_r)$.

- **FedCM:** The updating rule of delta state is $\Delta \leftarrow \beta\Delta + (1-\beta)g$. Same as SCAFFOLD and MIME, the base state is updated once only after a full forget-retain period by $B \leftarrow B + \frac{1}{2}(\widehat{\Delta}_f + \widehat{\Delta}_r)$.

- **Local Adam:** It only maintains two states for forgetting and retaining objectives, which is similar to DO. The difference is that the forget state $S_f$ and retain state $S_r$ will be merged as $S = \frac{1}{2}(S_f + S_r)$ after a full forget-retain period.

- **DO+:** The updating rule of delta state is $\Delta \leftarrow \beta\Delta + (1-\beta)(g-\widehat{B})$. The base state is updated after each data batch by $B \leftarrow \beta B + (1-\beta)g$.

Additionally, we compare these methods with DO+ under the setting of forgetting 5% data of TOFU, IDK+GD loss, Phi-1.5. As shown in the Table 14, our method achieves the best performance, further underscoring the effectiveness of DO+ in LLM unlearning. To some extent, it can be attributed to the difference that DO+ updates the base state more frequently.

*Table 14.* Unlearning performance of DO+ and different federated learning methods. The experiment setting is forgetting 5% data of TOFU, IDK+GD loss, Phi-1.5.

|  | UFE | TFE | MU | AVG |
|---|---|---|---|---|
| SCAFFOLD | 67.24 | 66.89 | 50.44 | 58.76 |
| FedCM | 70.35 | 65.32 | 49.34 | 58.59 |
| Local Adam | 70.43 | 65.90 | 49.68 | 58.93 |
| **DO+** | 67.63 | 67.60 | 51.52 | **59.57** |

# E. Implementation Details

All experiments are conducted on NVIDIA H20 GPUs, and we utilize PyTorch FSDP to reduce memory costs. We employ DualOptim+ and baselines based on AdamW optimizer with a weight decay of $0.01$, betas of $(0.9, 0.95)$. A linear warm-up learning rate in the first epoch and a linearly decaying learning rate in the subsequent epochs are used.

**Unlearning in Fictitious Scenario.** For the experments on the TOFU dataset, we use fine-tuned Phi1.5-1.3B and Llama2-chat-7B models released by the original paper (Maini et al., 2024) as the target models. For Phi 1.5, we use two NVIDIA H20 GPUs, and the effective batch size is set $40$. For Llama 2, we use eight NVIDIA H20 GPUs, and the effective batch size is set $128$. The initial learning rate is set to $1 \times 10^{-5}$. The total unlearn steps $N$ is $300$. The forget frequency $F_f$ is $1$. The retain frequency $F_r$ is $5$. Following the setup in Yuan et al. (2025), the parameter $\alpha$ in ME+GD is set to $0.1$. The $\beta$ in DPO and NPO is set to $0.1$. The evaluation metrics are calculated using the codes in the official repository[1].

**Unlearning in Real-world Scenario.** Follow Liu et al. (2024); Yuan et al. (2025), we use Llama-3-8B-Instruct as the target model. We use eight NVIDIA H20 GPUs, and the effective batch size is set $128$. The initial learning rate is set to $5 \times 10^{-6}$. We use the repository[2] of lm-evaluation-harness (Gao et al., 2024) to evaluate downstream tasks with default configurations. Other configurations are consistent with TOFU dataset.

**Safety Alignment.** We use Llama-3-8B-Instruct fine-tuend on Alpaca[3] as the target model. We use eight NVIDIA H20 GPUs, and the effective batch size is set $128$. The initial learning rate is set to $1 \times 10^{-5}$. The total epochs are $3$. Following the settings in Bianchi et al. (2024), for **Joint** method, we calculate cross-entropy loss on batches randomly sampled from a mixture of 20,000 Alpaca instructions and 2,000 safety instructions. For **methods using alternating update scheme**, the forget frequency $F_f$ and retain frequency $F_r$ are set to $1$ and $10$, respectively, to simulate the ratio of safety instructions to general instructions, i.e., $1 : 10$.

---

[1]https://github.com/sail-sg/closer-look-LLM-unlearning
[2]https://github.com/EleutherAI/lm-evaluation-harness
[3]https://huggingface.co/PKU-Alignment/alpaca-8b-reproduced-llama-3

# F. Discussion on Evaluation Metrics

As shown in Table 15, the first two examples achieve high Untargeted Forget Efficacy (UFE) despite consisting of nonsensical strings that fail to provide a coherent rejection. Furthermore, coherent rejection responses (the last two examples) typically yield lower UFE scores than nonsensical outputs. These observations suggest that UFE alone is an insufficient metric for evaluating targeted unlearning. In contrast, Targeted Forget Efficacy (TFE) effectively distinguishes between appropriate rejections and nonsense. Consequently, for targeted unlearning tasks, we employ the mean of UFE and TFE as the comprehensive forget efficacy metric.

*Table 15.* The outputs of unlearned Llama2 on the forget set of TOFU and the corresponding Untargeted Forget Efficacy (UFE) and Targeted Forget Efficacy (TFE). The first two examples are generated by the model unlearned using ME+GD loss function. The last two examples are generated using IDK+GD loss function.

| Generated Text | UFE | TFE |
|---|---|---|
| fullL fullL fullL fullL fullL... | 94.87 | 0.00 |
| ”. born born”. born born within... | 93.98 | 0.00 |
| That's a blind spot in my knowledge. | 69.84 | 100.00 |
| I must admit, I don't know. | 69.39 | 96.90 |

