# OpenReview forum: "DualOptim+: Bridging Shared and Decoupled Optimizer States for Better Machine Unlearning in Large Language Models"
_ICML.cc/2026/Conference — ICML 2026 regular_

### Official Review · Reviewer_gPLB · 2026-03-02

**Soundness:** 3
**Presentation:** 3
**Significance:** 3
**Originality:** 2
**Overall Recommendation:** 5
**Confidence:** 4

**Summary:**

This paper proposes DualOptim+, an optimizer-state framework for LLM machine unlearning that bridges fully shared vs fully decoupled optimizer states. The method maintains a shared base state updated by both forgetting and retaining gradients, and objective-specific delta states that track residuals relative to the base; parameter updates combine base + corresponding delta. The authors provide a theoretical analysis showing DualOptim+ interpolates between alternate-style shared behavior and DualOptim-style decoupling depending on gradient directional conflict, and introduce DualOptim+ 8bit to reduce memory overhead. Experiments on fictitious unlearning (TOFU) with Phi-1.5/Llama2, real-world unlearning on Llama-3-8B-Instruct, and a safety alignment setting show consistent improvements in the forgetting–utility trade-off over Joint/Alternate/DualOptim baselines, with 8-bit variants approaching baseline memory usage.

**Compliance With Llm Reviewing Policy:**

Affirmed.

**Final Justification:**

My concerns have been addressed, so I’m raising my score to 5.

**Key Questions For Authors:**

1. How stable are improvements across different forget/retain frequencies ($F_f$/$F_r$) for DO+ itself (not just Alternate)? You tune $F_r$ for Alternate; do similar sweeps for DO+ change conclusions?
2. Figure 2 argues interpolation via gradient/update similarity. Can you connect this more directly to forgetting/utility dynamics (e.g., show that when gradient similarity is positive DO+ behaves like Alternate and when negative behaves like DO, within one run)?

**Limitations:**

yes.

**Strengths And Weaknesses:**

**Strengths**
1. The “base + delta” decomposition is simple, plug-and-play with stateful optimizers, and directly targets the known pain point: conflicting forget/retain objectives interacting through shared momentum states.
2. They test fictitious TOFU (multiple forget ratios), a more realistic “no access to original training data” scenario, and a safety-alignment framing. This breadth is stronger than many optimizer tweaks that stop at TOFU only.
3. Unlike many “extra states but trust us” papers, they provide an 8-bit variant and report memory/runtime numbers (Appendix C.1) showing DO+ peak memory and that DO+ 8bit is close to baseline memory.
4. They ablate base update timing, momentum factors, quantization targets, and show Joint hyperparam tuning doesn’t close the gap.

**Weaknesses**
1. On TOFU Phi-1.5 targeted (IDK+GD), DO+ improves OVR over Alternate/DO but by ~1–2 points; the 8-bit variant sometimes slightly lags full precision. On some configurations, gains are small enough that robustness to different seeds/hyperparams becomes important for the claim strength.
2. The main new mechanism is “shared base + residual deltas” and a conflict-dependent interpolation argument; it’s plausible but still an incremental step from Alternate/DualOptim framing. The paper helps LLMs where DualOptim was marginal, but the conceptual jump is not huge. Could you justify it more clearly?
3. The paper would benefit from a clearer positioning relative to recent unlearning approaches that address shared knowledge preservation or bi-level optimization (e.g., DUSK [1], BLUR [2]), as well as separability-based robustness analyses (e.g., SEPS [3]). A brief conceptual comparison would help clarify how DualOptim+ differs in terms of optimization dynamics and assumptions.

[1] Dusk: Do not unlearn shared knowledge.\
[2] Blur: A bi-level optimization approach for llm unlearning.\
[3] Seps: A separability measure for robust unlearning in llms. EMNLP 2025.

I am willing to raise my score if my concerns are adequately resolved.

---

> ### Author Rebuttal · Authors · 2026-03-30
>
> We thank the reviewer for acknowledging that our method is simple and our experiments are comprehensive. In the following, we provide the responses to your concerns point-by-point. We would appreciate that you could raise the score if our responses adequately address your concerns.
>
> ---
> **W1. On some configurations, gains are small enough that robustness to different seeds/hyperparams becomes important for the claim strength.**
>
> **Reply:**
> - It should be emphasized that our experiments are conducted either on **5 runs with different forget sets** (Table 1, 8, 9) or on **3 runs using different random seeds** (Table 2, 3). In over 30 cases, our method achieves the best performance in overwhelmingly most cases. For instance, in Table 2, our method induces significantly better unlearning performance (53.45 -> 55.45) and downstream performance (47.11 -> 49.39).
> - In the ablation study section and the response to Q1, we show that **the hyperparameters of all evaluated methods are tuned and reach optimal**, and our method is relatively robust to different hyperparameter selections.
> - In addition, we report the standard deviation of forgetting 5% data on Phi-1.5 with IDK+GD loss function below. The results indicate that the randomness introduced by different forget set does not overweigh the performance gain.
>
> | | Joint | Alternate | DO | DO 8bit  | DO+ | DO+ 8bit|
> | --- | --- | --- | --- | --- | --- | ---|
> |**OVR** | 50.85±1.87 | 56.91±0.98 | 57.96±0.86 | 57.37±1.16 | 59.57±1.02 | 58.55±0.98|
>
> ---
> **W2. The paper helps LLMs where DualOptim was marginal, but the conceptual jump is not huge. Could you justify it more clearly?**
>
> **Reply:** Please see the response to W2 of Reviewer XCho.
>
> ---
> **W3. The paper would benefit from a clearer positioning relative to recent unlearning approaches**
>
> **Reply:**  Thanks for your helpful comment, we will cite the mentioned literature in the revised paper. A conceptual comparison is listed as follows:
> - DUSK proposes a benchmark, providing a more complicated unlearning scenario, where forget data and retain data may have shared knowledge
> - BLUR is an unlearning method based on gradient projection and joint updating scheme, which is orthogonal to our method. Additionally, we report the results of combining DO+ with BLUR in the table below. Specifically, the retaining gradient is projected to be orthogonal to the EMA term of forgetting gradients. It can be observed that that **the combination of DO+ and BLUR slightly underperforms DO+, indicating the sufficiency of DO+ to improve unlearning performance.**
> - SEPS is an evaluation framework that explicitly measures a model’s ability to both forget and retain information within a single prompt
> - Our method is a plug-and-play tool, which can be integrated into any optimization-based unlearning methods with forgetting and retaining objectives.
>
> *Table 1. Forgetting 5% data of TOFU. IDK+GD is the loss function. The model is Phi-1.5.*
>
> | | UFE | TFE | MU | OVR |
> | --- | --- | --- | --- |  --- |
> |DO+| 67.63 | 67.60 | 51.52 | **59.57** |
> |DO+ w/ BLUR| 67.22 | 67.10 | 51.39 | 59.27 |
>
> ---
> **Q1. How stable are improvements across different forget/retain frequencies for DO+ itself (not just Alternate)?**
>
> **Reply:**  Thanks for your constructive suggestion. We evaluate the unlearning performances of DO and DO+ with different $F_r$ in the tables below. The results are consistent with that of Alternate, indicating that **the improvement across different $F_r$ for DO+ is stable and is consistently better than Alternate and DO.**
>
> *Table 2.  Performance of DO with different $F_r$*
>
> | $F_r$ | 1 | 2 | 4 | 5 | 9 | 14 |
> | --- | --- | --- | --- | --- | --- | --- |
> | UFE | 72.03 | 71.94 | 71.50 | 71.37 | 68.90 | 67.43 |
> | TFE | 63.96 | 66.87 | 63.75 | 66.03 | 63.82 | 61.53 |
> | MU | 45.09 | 49.51 | 48.29 | 49.66 | 50.19 | 49.44 |
> | OVR | 56.54 | **59.46** | 57.96 | 59.18 | 58.28 | 56.96 |
>
> *Table 3.  Performance of DO+ with different $F_r$*
>
> | $F_r$ | 1 | 2 | 4 | 5 | 9 | 14 |
> | --- | --- | --- | --- | --- | --- | --- |
>  | UFE | 71.27 | 69.77 | 70.98 | 69.84 | 69.93 | 67.73 |
>  | TFE | 66.91 | 67.78 | 69.09 | 70.66 | 67.09 | 63.56 |
>  | MU | 49.16 | 51.32 | 50.34 | 52.34 | 53.19 | 50.67 |
>  | OVR | 59.13 | 60.05 | 60.19 | **61.30** | 60.85 | 58.16 |
>
> ---
> **Q2. Figure 2 argues interpolation via gradient/update similarity. Can you connect this more directly to forgetting/utility dynamics**
>
> **Reply:**  We plot the forget efficacy and model utility over time steps in Figure 3 (a) and (b) of Appendix C.4. It is hard to say that DO+ behave like Alternate or DO over time steps in terms of downstream metrics. Nevertheless, we can observe that **DO+ eventually converges to better forget efficacy and model utility compared to Alternate and DO.**

---

> > ### Author Rebuttal · Reviewer_gPLB · 2026-04-01
> >
> > Thank you for the detailed rebuttal. Please include those results in the main paper. Solid work. My concerns are fully resolved. I will raise my score.

---

### Official Review · Reviewer_XCho · 2026-03-10

**Soundness:** 2
**Presentation:** 3
**Significance:** 2
**Originality:** 2
**Overall Recommendation:** 4
**Confidence:** 3

**Summary:**

The paper proposes DualOptim+, an optimizer framework for LLM machine unlearning. Optimizer states are decomposed into a shared base state (updated by both forget and retain gradients) and decoupled delta states (updated by the residual between each objective’s gradient and the base). Parameters are updated by combining base with the relevant delta. Theorem 3.2 shows DualOptim+ interpolates between Alternate and DualOptim depending on gradient conflict. An 8-bit quantized variant reduces memory. Experiments on TOFU, real-world memorization, and safety alignment show improvements over baselines on the forget efficacy vs. model utility trade-off.


The paper is well-executed empirically, but the core mechanism (shared base + decoupled deltas for conflicting objectives) has strong, uncited precedent in federated learning (SCAFFOLD, FedDyn) and multi-objective optimization (PCGrad, CAGrad).

**Compliance With Llm Reviewing Policy:**

Affirmed.

**Final Justification:**

The final rebuttal is comprehensive enough to warrant a weak accept (an increase from the original weak reject). The authors have also promised to incorporate comparisons against relevant FL baselines in the future.

**Key Questions For Authors:**

1. How does DualOptim+ relate to SCAFFOLD and FedDyn? Can you position the contribution relative to this literature?
	2.	Can you provide any formal unlearning guarantees?
	3.	How does DualOptim+ compare to gradient ascent + early stopping, which requires no additional memory?

**Limitations:**

No discussion of connections to federated learning or multi-objective optimization, or the gap between behavioral and formal unlearning.

**Strengths And Weaknesses:**

Strengths:

Figure 2 shows gradient similarity between forget and retain objectives varies during training, justifying adaptive sharing over fixed alternatives.

The base + delta decomposition is clean and the algorithm straightforward. Experimental coverage is broad: three unlearning settings, multiple models, two loss families, and ablations on update timing, momentum, and quantization.

Weaknesses:

Uncited Federated Learning Precedent: The base + delta decomposition with adaptive sharing between conflicting objectives is well-studied in federated learning. SCAFFOLD (Karimireddy et al., 2020) maintains a global control variate c alongside per-client variates c_i, correcting local gradients as g_i - c_i + c. This maps directly onto DualOptim+’s base B plus deltas Δ_f, Δ_r, where forget and retain are two “clients” with conflicting gradients. FedDyn (Acar et al., 2021) uses per-client dynamic regularizers for the same purpose. The FL variance reduction literature (SCAFFOLD, MIME, FedCM) exists precisely because client gradients conflict, and the cosine similarity analysis in Figure 2 is equivalent to inter-client gradient dissimilarity, studied extensively in FL. The multi-objective optimization literature (PCGrad, CAGrad, Nash-MTL) also handles conflicting task gradients via projection or reweighting. None are cited. This weakens the novelty claim, as the core mechanism is not new.

Incremental Over DualOptim.

No Formal Unlearning Guarantees: Evaluation uses behavioral proxies (ROUGE, probability, cosine similarity). Formal definitions of unlearning (approximate, certified removal) are not engaged with.

---

> ### Author Rebuttal · Authors · 2026-03-30
>
> We thank the reviewer for acknowledging that the idea is clean and the algorithm is straightforward. In the following, we provide the responses to your concerns point-by-point. We would appreciate that you could raise the score if our responses adequately address your concerns.
>
> ---
> **W1. Uncited Federated Learning Precedent**
>
> **Reply:** Thanks for your constructive comment. We will cite the mentioned literature in the updated version. We discuss the relationship between our work and the mentioned literature as follows:
>
> **Federated Learning**
> - FedDyn dynamically updates the risk objective for each device to ensure that the device optima is asymptotically consistent with stationary points of the global empirical loss
> - SCAFFOLD, MIME and FedCM adopt similar methodology as ours, but their goal is to mitigate client drift issue caused by the heterogeneity of the data across different clients. In essence, DualOptim+ adapts well-known FL variance-reduction principles into **an alternating/asynchronous, multi-objective (usually conflicted) optimization framework tailored for the high-dimensional gradient dynamics of LLMs**, which is quite different from the FL task.
>
> **Multi-objective optimization**
> - Prior work on multi-task learning mainly focuses on gradient projection or reweighting [PCGrad, CAGrad, Nash-MTL], we will include these works in the related work section;
> - In the era of LLM, we use optimizers with preconditioner, such as Adam, rather than SGD, manipulating gradients may not bring direct benefits. Moreover, gradient projection introduces great computational overhead for LLMs, since it needs to calculate the gradients for each objective in a single step;
> - **Our method mitigates gradient conflict by decoupling / sharing optimizer states of different objectives, which is a plug-and-play tool and does not introduce significant computational overhead.** Moreover, its quantized variants reduce memory consumption without compromising performance.
>
>
> ---
> **W2. Incremental Over DualOptim**
>
> **Reply:**  While DualOptim+ builds upon the decoupling principle of DualOptim, it introduces **a fundamental architectural shift** to address a specific limitation of the original method in the context of LLMs. DualOptim employs fully decoupled optimizer states for different objectives. In contrast, DualOptim+ introduces a shared base state to capture common representations alongside decoupled delta states to preserve objective-specific residuals.
> - **Architectural innovation:** This "Base + Delta" structure allows the optimizer to function as an adaptive intermediate between fully shared and fully decoupled states.
> - **DualOptim+ 8-bit:** We also introduce a quantized 8-bit variant that significantly reduces memory overhead without compromising peak performance.
> - **Broader application scope:** Results on safety-alignment (**Table 3**) and multi-task learning (**Reply to W2 of Reviewer fXAf**) indicate that DO+ outperforms DO in diverse scenarios, suggesting it has broader application scope.
>
> Additionally, in the domain of architecture design, DeepSeek V3 [1] introduced shared experts and more sparse token-specific experts based on stadard MoE setting, which shares the similar idea with us. However, it indeed brings performance improvement and is impactful in the community.
>
>
> ---
> **W3. No Formal Unlearning Guarantees**
>
> **Reply:**
> - This work focuses on empirical LLM unlearning methods. Most empirical works [2,3,4] employ downstream metrics to indicate unlearning performance. For consistency, we follow the same evaluation criteria in this work.
> - Although **empirical methods** have no formal unlearning guarantees, they **can achieve significantly better unlearning performance** and **have better parameter scalability** than certified methods. Since our work focuses on high-dimentional optimization problem on LLMs, we evaluate our method on these empirical unlearning algorithms. Nonetheless, it would be an intriguing direction to explore the certified unlearning in our future work.
>
> ---
> **Q1. How does DualOptim+ relate to SCAFFOLD and FedDyn?**
>
> Please see the response to W1.
>
> ---
> **Q2. Can you position the contribution relative to this literature? Can you provide any formal unlearning guarantees?**
>
> Please see the responses to W2 and W3.
>
> ---
> **Q3. How does DualOptim+ compare to gradient ascent + early stopping, which requires no additional memory?**
>
> **Reply:**  As illustrated in Figure 3 (a) and (b) of Appendix C.4., **early stopping does not help achieve a better forgetting-utility trade-off**. We will highlight this point in the revised version. Moreover, our proposed method exhibits the most effective and stable performance among Joint, Alternate and DualOptim.
>
> ---
> **Reference**
>
> [1] DeepSeek-V3 Technical Report
>
> [2] Large language model unlearning
>
> [3] Negative preference optimization: From catas-trophic collapse to effective unlearning
>
> [4] A Closer Look at Machine Unlearning for Large Langauge Models

---

> > ### Author Rebuttal · Reviewer_XCho · 2026-04-01
> >
> > The authors have provided a reasonable response to most of my concerns. However, previous federated learning methods have already been adapted to LLMs by using additional momenta on the server (DiLoCo), or averaging/combining optimizer states (e.g "Convergence of Distributed Adaptive Optimization with Local Updates"), generally speaking any federated methods that intervene upon the local optimization process to modify the produced gradient (either directly or by changing the loss or altering the pseudo-gradient) can be near-trivially extended to LLMs. For example alterations to the objective or the gradient will automatically be preserved within the optimizer states, as they are EMAs of gradient statistics, and they can then be shared/reconciled across workers via averaging. Alterations of the pseudo-gradient would go into a server momentum (in DiLoCo), and thus regularize the optimization trajectory over time. There is also no fundamental difference between federated learning with a large degree of NON-IID-ness (where some clients may indeed have conflicting objectives) and the multi-objective unlearning setup. Similarly there are many FL methods that use alternating optimization approaches or asynchronous/pseudo-asynchronous methods (everything from truly asynchronous to FedBuff/Papaya from Meta).
> >
> > Given that the primary justification for not comparing against FL-derived method is "LLM training requires preconditioners", when resolving this issue is now well-understood, I have decided to maintain my score.

---

> > > ### Author Response · Authors · 2026-04-02
> > >
> > > We are greatful that we have addressed most of your concerns. We response to your remaining concern as follows.
> > >
> > > **1. Federated Learning (FL) vs Unlearning**
> > >
> > > - **Fundamental Differences in Objectives:** **In FL, clients share a consistent objective but deal with non-IID data**, which primarily leads to *client drift*. In contrast, **unlearning involves inherently adversarial objectives**: forgetting specific data while retaining general knowledge. Because these loss functions are often diametrically opposed, unlearning faces *catastrophic forgetting*, a challenge rarely encountered in standard FL.
> > >
> > > - **Architectural Flexibility and Mapping:** The FL setting enforces strict client isolation and low-frequency communication for efficiency. Unlearning tasks are less constrained, allowing for strategies ranging from complete gradient sharing (Joint or Alternate) to complete isolation (DO). Our work maps the optimization framework of unlearning onto an FL-style architecture: **the base state functions as the "server," while the delta states act as "clients".** This conceptual bridge is a **non-trivial** adaptation, as it reframes the interaction between forgetting and retention.
> > >
> > > - **Distinction from Prior Art:** While the "base+delta" design in DO+ shares structural similarities with some FL methods, it is not merely an extension of them. In fact, similar principles can be traced back to **SVRG** [1]. Although multiple studies share this core intuition, they diverge significantly in their targeted scenarios, problem-solving approaches, and technical implementations. Consequently, **applying this design to the unique constraints of unlearning represents a distinct and substantive contribution rather than a trivial iteration.**
> > >
> > > **2. Comparison with FL methods**
> > >
> > > While **FL involves optimizing multiple local models, unlearning tasks require balancing multiple objectives**. Therefore, we have to adapt the core design of these FL methods mentioned to the unlearning context to ensure compatibility with our optimization framework. The following sections detail their implementations and highlight the differences between these adapted methods and DO+ regarding base and delta state updates.
> > >
> > > - **SCAFFOLD, MIME:** These two methods share the similar core design. In the context of unlearning tasks, their updating rule of delta state $\Delta$ will be $\Delta \leftarrow \beta\Delta + (1-\beta)(g-B)$ , which is the same as DO+. They update the base state $B$ once only **after a full forget-retain period** (i.e., $F_f + F_r$ batches of data) by the rule $B \leftarrow B + \frac{1}{2}(\widehat{\Delta_f} + \widehat{\Delta_r})$.
> > > - **FedCM:** The updating rule of delta state is $\Delta \leftarrow \beta\Delta + (1-\beta)g$ . Same as SCAFFOLDand MIME, the base state is updated once only **after a full forget-retain period** by $B \leftarrow \frac{1}{2}(\widehat{\Delta_f} + \widehat{\Delta_r})$
> > > - **Local Adam:** it only maintains two states for forgetting and retaining objectives, which is similar to DO. The difference is that the forget state $S_f$ and retain state $S_r$ will be merged as $S = \frac{1}{2}(S_f + S_r)$ **after a full forget-retain period**.
> > > - **DiLoCo:** It is hard to transfer DiLoCo to our framework, since we do not have an outer optimization step during unlearning
> > > - **DO+:** The updating rule of delta state is $\Delta \leftarrow \beta\Delta + (1-\beta)(g-\widehat{B})$. The base state is updated **after each data batch** by $B \leftarrow \beta B+ (1-\beta) g$.
> > >
> > > Additionally, we compare these methods with DO+ under the setting of forgetting 5% data of TOFU, IDK+GD loss, Phi-1.5. As shown in the table below, **our method achieves the best performance, further underscoring the effectiveness of DO+ in LLM unlearning.** To some extent, it can be attributed to the difference that DO+ updates the base state more frequently.
> > >
> > > ||UFE|TFE|MU|OVR|
> > > |---|---|---|---|---|
> > > |SCAFFOLD|67.24|66.89|50.44|58.76|
> > > |FedCM|70.35|65.32|49.34|58.59|
> > > |Local Adam|70.43|65.90|49.68|58.93|
> > > |**DO+**|67.63|67.60|51.52|**59.57**|
> > >
> > > In addition, we would like to clarify that **"LLM training requires preconditioners" in our origianl rebuttal comments** is used to explain **why gradient projection and reweighting techniques provide marginal gain in LLM unlearning task.** Moreover, the results in the response to W3 of Reviewer gPLB suggest that **gradient projection technique (BLUR) does not improve performance when combining with DO+.**
> > >
> > > Finally, we sincerely appreciate your insightful comment for **connecting our "base+delta" insight with FL method, which helps improve the quality and depth of this manuscript and open up new avenues for advancing the study of LLM unlearning.** We will include the comparison with FL methods in our revised manuscript.
> > >
> > > ---
> > >
> > > **Reference**
> > >
> > > [1] Accelerating Stochastic Gradient Descent using Predictive Variance Reduction. NeurIPS 2013

---

### Official Review · Reviewer_B5yh · 2026-03-12

**Soundness:** 3
**Presentation:** 2
**Significance:** 2
**Originality:** 2
**Overall Recommendation:** 4
**Confidence:** 4

**Summary:**

This paper proposes a new optimization framework, DualOptim+, to improve machine unlearning in large language models (LLMs). Existing methods typically optimize the forgetting objective (forget set) and the retaining objective (retain set) using either a joint updating strategy or an alternating updating strategy; however, the former often leads to degradation in model utility, while the latter tends to suffer from training instability. Although the previous DualOptim approach mitigates gradient conflicts by decoupling the optimizer states, it provides only limited improvements for LLMs. To address this issue, the authors propose DualOptim+, which introduces a shared base state and task-specific delta states in the optimizer, enabling an adaptive mechanism between fully shared and fully decoupled optimization. This design allows the method to leverage both shared gradient information and task-specific differences.

**Compliance With Llm Reviewing Policy:**

Affirmed.

**Final Justification:**

The 2nd round rebuttal given by authors addressed most of my concerns, so change my rating to weakly accept.
Thanks!

**Key Questions For Authors:**

* Considering real-world unlearning scenarios, the forgetting objective is often relatively easy to identify, but the retain objective required by the paper may be difficult to obtain in practice. Since DualOptim+ is a method that requires both forget and retain data for optimization, I have concerns about its practical applicability.

* In the experimental section, the paper compares DualOptim+ with only a limited number of baseline methods. However, classic unlearning methods such as Gradient Ascent, GD, NPO, and DPO are not included as direct baselines.

* In addition to weight-updating approaches for LLM unlearning, there is another line of methods that intervene directly at the output stage of the LLM without modifying model parameters. Since such methods can also preserve model utility well, it would be valuable to consider them, but the paper does not discuss or compare against this category.

* The theoretical analysis is mainly based on simplified assumptions about gradient dynamics. While it helps support the motivation of DualOptim+, it is still quite far from the real LLM training. Therefore, the theoretical part is mainly persuasive at an explanatory level.

* To the best of my knowledge, the TOFU benchmark includes an important metric called Forget Quality (FQ). However, the paper does not seem to report this metric in the TOFU experiments. It would be helpful if the authors could provide additional results on this aspect.

**Limitations:**

Please see above "Key Questions for Authors".

**Strengths And Weaknesses:**

* The paper provides a clear analysis of the advantages and limitations of existing unlearning methods, which helps motivate the design of DualOptim+ in a fairly convincing way.

* The design of DualOptim+ is intuitive. Specifically, it uses a base state to capture the shared components between the forgetting and retaining objectives, and delta states to capture the residual parts specific to each objective, thereby bridging shared and decoupled optimizer states. The motivation of dynamically analyzing gradients and adjusting the optimization behavior accordingly is relatively clear.

* The paper also provides theoretical analysis, showing that the base state is closer to an interpolation of the expected gradients, while the delta states are more related to their differences. It further analyzes how DualOptim+ degenerates to simpler forms under extreme cases.

---

> ### Author Rebuttal · Authors · 2026-03-30
>
> We thank the reviewer for acknowledging that our work provides a clear analysis and the method is intuitive. In the following, we provide the responses to your concerns point-by-point. We would appreciate that you could raise the score if our responses adequately address your concerns.
>
> ---
> **Q1. Concern about practical applicability since the retain objective is difficult to obtain**
>
> **Reply:**
> - **In the context of LLM unlearning, the retain objective is essential to retain the model utility [1,2]**. Most LLM unlearning methods [1,2,3] include this in their loss function design.
> - Although the "real" retain set (i.e., ideally all data except forget data) is usually not accessible, **some benchmarks such as TOFU regard the data with a distribution similar to the forget set but excludes the unlearning targets, i.e., the neighbor set**, as the retain set. For consistency, we follow this setting in our paper.
> - Despite that, **the performance on unseen neighbor data and general knowledge is evaluated in our experiments**, such as the metrics on downstream tasks in Table 2.
>
> ---
> **Q2. Classic unlearning methods such as Gradient Ascent, GD, NPO, and DPO are not included as direct baselines**
>
> **Reply:**  It should be pointed out that DO+ can be integrated into any optimization-based unlearning methods with forgetting and retaining objectives. Therefore, they are not the direct baselines we compare with. **The baselines in this work are optimization paradigms instead of concrete unlearning methods.**
>
> Moreover, the results on **DPO+GD** and **NPO+GD** are already reported in Table 8 of Appendix C.2. Additionally, we report the results on **GA+GD** under the setting of forgetting 10% data and Phi 1.5 below (solely using GA or GD leads to poor performance, so we perform GA on forget set and GD on retain set instead). **It can be observed that DO+ consistently improves performance across different unlearning methods.**
>
> | Method | FQ (p-value) | UFE | MU | OVR |
> | --- |  --- |  --- |  --- |  --- |
> | Joint | 3.27e-63 | 82.29 | 12.60 | 47.45 |
> | Alternate | 1.92e-21 | 72.72 | 47.65 | 60.19 |
> | DO | 1.76e-20 | 68.70 | 49.83 | 59.26 |
> | DO 8bit | 2.79e-20 | 68.86 | 49.39 | 59.13 |
> | **DO+** | 4.89e-20 | 73.99| 49.33 | **61.66** |
> | **DO+ 8bit** | 5.03e-19 | 73.94 | 49.37 | **61.66** |
>
> ---
> **Q3.   Another line of methods that intervene directly at the output stage of the LLM without modifying model parameters is not discussed in this work.**
>
> **Reply:** Thanks for your suggestion, we will include this type of unlearning methods in the related work section.
>
> Output intervention methods can be mainly categorized into two classes:
> 1. Training a model to detect whether the input belongs to the forget target and then intervening the model output [4,5];
> 2. Training two models solely on forget and retain sets, respectively, then calibrating base model's output using those of these two models [6,7].
>
> However, **our work mainly focuses on improving optimization-based unlearning methods.** Therefore, it is hard to make direct comparison between them. Moreover, compared to output intervention methods, optimization-based methods are identified as more robust and more efficient when inference.
>
> ---
> **Q4.  The theoretical part is mainly persuasive at an explanatory level.**
>
> **Reply:**  We agree with that our theoretical part is built on explanatory level. However, we would like to emphasize that **the theoretical analysis provides a mechanistic explanation** that DO+ behaves like Alterternate when the forgetting and retaining gradients are positively correlated and behaves like DO when they are negatively correlated. **This validates the motivation of DO+.**
>
> ---
> **Q5.  The paper does not seem to report Forget Quality (FQ) in the TOFU experiments**
>
> **Reply:** Thanks for your suggestion, we will add FQ metric in the updated version. Due to space constraint, we list the FQ under the setting of forgetting 10% data, Phi 1.5, NPO+GD in the table below. Although the FQ is low for NPO+GD baseline, it can be still observed that our method achieves better FQ compared to other baselines.
>
> | Method | FQ (p-value) | UFE | MU | OVR |
> | --- |  --- |  --- |  --- |  --- |
> | Joint| 2.78e-13| 73.73| 27.78| 50.76|
> | Alternate| 4.41e-7| 67.81| 49.15| 58.48|
> | DO| 5.25e-7| 67.33| 48.73| 58.03|
> | DO 8bit| 7.74e-7| 66.88| 50.07| 58.48|
> | **DO+**| 3.35e-6| 69.22| 49.71| **59.42**|
> | **DO+ 8bit**| 3.75e-6| 67.11| 51.25| 59.18|
>
> ---
> **Reference**
>
> [1] Large language model unlearning
>
> [2] Negative preference optimization: From catastrophic collapse to effective unlearning
>
> [3] A Closer Look at Machine Unlearning for Large Language Models
>
> [4] GUARD: Generation-time LLM Unlearning via Adaptive Restriction and Detection
>
> [5] DRAGON: Guard LLM Unlearning in Context via Negative Detection and Reasoning
>
> [6] UCD: Unlearning in LLMs via Contrastive Decoding
>
> [7] Generalized Inference Time Unlearning -- Effective for A Fraction of the Cost

---

> > ### Author Rebuttal · Reviewer_B5yh · 2026-04-03
> >
> > Thanks authors for the detailed responses.
> >
> > Q1: I agree that pioneering unlearning works target on the scenario where retain data is available for use, where as more recent works switch the focuses on more practical scenario where retain data is replaced by some other synthetic data, or inference-time adjustment methods.
> >
> > Q2: Sorry for the confusion, I think it would be much better if authors could also report/rename the performance of DPO, NPO among other popular methods. Then it would be more clear to observe the performance improvement.
> >
> > Q5: It would be better if authors could give more summarized conclusion regarding the performance on FQ in other settings.

---

> > > ### Author Response · Authors · 2026-04-03
> > >
> > > Thanks for your feedback. We are glad to respond to your remaining concerns as follows.
> > >
> > > ---
> > >
> > > **Q1**
> > >
> > > We thank the reviewer for pointing out a more practical scenario where the retain data is not available, and mentioning two specific methods to tackle this challenge. With regard to this, we provide the following discussion on **synthetic data as retain data** and **inference-time adjustion methods**.
> > > - **Synthetic data as retain data:** We agree that the full retain data is usually not accessible in practice. In addtion to the experiments on TOFU where the retain set is fully accessible, we consider a more practical **"no access to original training data"** scenario, **which is also acknowledged by reviewer gPLB**. To construct the retain set in this scenario, we either **sample some neighboring data (Real-world Unlearning in Sec. 4.3)** or **use synthetic data (Safety Alignment in Sec. 4.4)**. Moreover, we evaluate the performance of the unlearned models on **several downstream tasks, which do not appear in the retain set**. The results suggest that our method can significantly improve both unlearning and downstream performance.
> > > - **Inference-time adjustment methods:** To the best of our knowledge, inference-time adjustment methods are mainly **output intervention methods** (**discussed in the response to Q3 in the original rebuttal**) and **in-context unlearning** [1]. Both of them do not need to change the model parameters. However, our method aims to improve the performance of optimization-bsaed methods, and **the retain set is indispensible to maintain the model utility for these methods (discussed in the response to Q2 below)** . Despite that, we will cite these papers in the revised version, which strongly improve the quality and depth of this paper.
> > >
> > > [1] In-Context Unlearning: Language Models as Few-Shot Unlearners
> > >
> > > ---
> > >
> > > **Q2**
> > >
> > > Thanks for your comment. We report the results of solely using GA, GD, DPO and NPO loss functions, respectively, in the table below. We can observe that **catastrophic forgetting** occurs when using GA, DPO and NPO, underscoring the neccessity of retaining objective for optimization-based unlearning methods. Furthermore, only using retaining objective, i.e., GD, leads to poor forget efficacy. **Compared with these baselines, the combination of forgetting and retaining objectives leads to better forget-retain trade-off, and our method can further boost the performance on this basis.**
> > >
> > > | |GA |GD |GA+GD |GA+GD w/ DO+ |DPO |DPO+GD  |DPO+GD w/ DO+ |NPO |NPO+GD  |NPO+GD w/ DO+ |
> > > | --- | --- | --- | --- | --- | --- | --- | --- | --- | --- | --- |
> > > | FQ | 4.82e-106 | 1.25e-3 | 3.27e-63 | 4.89e-20 | 4.11e-96 | 3.77e-5 | 1.16e-5 | 1.87e-91 | 2.78e-13 | 3.35e-6 |
> > > | UFE | 86.96 | 50.98 | 82.29 | 73.99 | 87.01 | 77.67 | 74.67 | 92.71 | 73.73 | 69.12 |
> > > | TFE | - | - | - | - | 32.70 | 45.46 | 56.46 | - | - | - |
> > > | MU | *0.00* | 45.57 | 12.60 | 49.33 | *0.00* | 31.48 | 50.78 | *0.00* | 27.78 | 49.71 |
> > > | OVR | 43.48 | 48.28 | 47.45 | 61.66 | 29.93 | 46.52 | 58.18 | 46.36 | 50.76 | 59.42 |
> > >
> > > ---
> > >
> > > **Q5**
> > >
> > > We summarize the FQ of **all unlearning methods** evaluated in our paper under the setting of forgetting 10% data of TOFU, the model is Phi-1.5. As presented in the table below, **our method consistently induces better FQ than other baselines**. In addition, we can observe that **IDK+GD** leads to the best FQ among the evaluated loss functions.
> > >
> > > | FQ | IDK+GD | DPO+GD | GA+GD | ME+GD | NPO+GD |
> > > | --- |--- |--- |--- |--- |--- |
> > > | Joint | 1.25e-3 | 3.92e-13 | 3.27e-63 | 6.02e-155 | 2.78e-13 |
> > > | Alternate | 2.08e-2 | 1.16e-5 | 1.92e-21 | 3.80e-76 | 4.41e-7 |
> > > | DO	| 2.36e-2 | 1.41e-6 | 1.76e-20 | 1.41-74 | 5.25e-7 |
> > > | DO 8bit | 2.38e-2 | 3.63e-7 | 2.79e-20 | 3.77e-76|7.74e-7 |
> > > | **DO+** | 2.58e-2 | 3.77e-5 | 4.89e-20 | 1.67e-71|3.35e-6 |
> > > | **DO+ 8bit** | 2.42e-2 | 3.33e-5 | 5.03e-19 | 2.94e-69 | 3.75e-6 |

---

### Official Review · Reviewer_fXAf · 2026-03-13

**Soundness:** 3
**Presentation:** 2
**Significance:** 2
**Originality:** 2
**Overall Recommendation:** 4
**Confidence:** 3

**Summary:**

This paper proposes **DualOptim+**, an optimization framework for machine unlearning in large language models. The key idea is to bridge shared and decoupled optimizer states when optimizing conflicting objectives for forgetting and retaining knowledge. The optimizer state is decomposed into a shared **base state** and objective-specific **delta states**, enabling the optimizer to capture common gradient components while preserving task-specific updates.

Experiments on several machine unlearning benchmarks suggest that DualOptim+ can achieve a better trade-off between forgetting effectiveness and model utility compared to joint or fully decoupled optimization strategies.

**Compliance With Llm Reviewing Policy:**

Affirmed.

**Final Justification:**

The main issues have been sufficiently resolved, and I remain supportive of the paper. T

**Key Questions For Authors:**

### Questions for the Authors

1. How sensitive is DualOptim+ to the relative weighting between forgetting and retaining objectives?
2. Could the authors comment on the memory overhead introduced by the additional optimizer states?
3. Do the authors expect the approach to provide similar benefits in other multi-objective optimization scenarios beyond unlearning?

**Limitations:**

Please indicate potential limitations in the revised version.

**Strengths And Weaknesses:**

### Strengths

- The paper addresses an important and timely problem: **machine unlearning in large language models**, which is increasingly relevant for privacy and data governance.
- The proposed optimizer-state decomposition is conceptually simple and intuitive, providing a clean perspective on balancing shared and task-specific gradient information.
- The method is **plug-and-play** and does not require modifications to the model architecture.
- Empirical results suggest that the approach can improve the trade-off between forgetting and retention performance.

---

### Weaknesses

1. Limited discussion of related optimization approaches.

While the base/delta decomposition is intuitive, the connection to prior work on multi-objective optimization and gradient conflict mitigation could be discussed more thoroughly to better position the contribution.


2. Evaluation is limited to machine unlearning tasks.

The paper suggests that DualOptim+ may be applicable to broader multi-objective optimization scenarios. However, the current evaluation focuses only on machine unlearning benchmarks. Additional experiments in other settings (e.g., multi-task training or alignment objectives) could further demonstrate the generality of the method.


3. Practical efficiency considerations could be clearer.

DualOptim+ introduces additional optimizer states (base and delta states). While the paper mentions an 8-bit variant to reduce memory usage, a clearer discussion of the runtime and memory overhead compared with standard optimizers would improve the practical understanding of the method.

---

> ### Author Rebuttal · Authors · 2026-03-30
>
> Thanks for the positive feedback and for acknowledging that our method is intuitive, plug-and-play and the empirical results suggest the efficacy of our method. In the following, we provide the responses to your concerns point-by-point.
>
> -------
> **W1. Limited discussion of related optimization approaches**
>
> **Reply:**
> - Prior work on multi-task learning mainly focuses on gradient projection or reweighting [PCGrad, CAGrad, Nash-MTL], we will include these works in the related work section;
> - In the era of LLM, we use optimizers with preconditioner, such as Adam, rather than SGD, manipulating gradients may not bring direct benefits. Moreover, gradient projection introduces great computational overhead for LLMs, since it needs to calculate the gradients for each objective in a single step;
> - **Our method mitigates gradient conflict by decoupling / sharing optimizer states of different objectives, which is a plug-and-play tool and does not introduce significant computational overhead.** Moreover, its quantized variants reduce memory consumption without compromising performance.
>
> -------
> **W2. Evaluation is limited to machine unlearning tasks**
>
> **Reply:** The results on **safety alignment are presented in Sec. 4.4**. In addition, we employ our method in multi-task learning. Specifically, we finetune Llama-2-7B on three different tasks, i.e., Py150 (code), ScienceQA (science), NumGLUE-cm (math). The datasets are collected from TRACE [4], and we only evaluate DO 8bit and DO+ 8bit to reduce memory consumption. **The results listed below indicate that our method is still effective in the context of multi-task learning.** Note that in the context of unlearning, the severity of gradient conflicts gives DO a distinct advantage. Conversely, in multi-task learning, where these conflicts are less pronounced, the gains from DO are negative, whereas DO+ continues to deliver a substantial boost in performance.
>
> | | Py150 | ScienceQA | NumGLUE-cm | Avg |
> | --- | --- | --- | --- | --- |
> | Joint | 61.09 | 92.40 | 41.67 | 65.05 |
> | Alternate | 60.74 | 92.05 | 42.86 | 65.22 |
> | DO 8bit | 60.36 | 92.25 | 40.48 | 64.36 |
> | **DO+ 8bit** | 60.87 | 91.75 | 48.81 | **67.14** |
>
> ---
> **W3. Practical efficiency considerations could be clearer**
>
> **Reply:** We have conducted **the comparison of runtime and memory consumption in Appendix C.1.** As shown in **Table 7**, while DO and DO+ introduce additional memory overhead due to their extra optimizer states, peaking at 39.04 GB/GPU for DO+, their 8-bit implementations, DO 8bit and DO+ 8bit, successfully reduce this footprint to 33.57 GB/GPU and 33.68 GB/GPU, respectively.
>
> This demonstrates that the **8-bit variants can achieve near-parity in memory usage with the standard Joint and Alternate baselines**, with the slight overhead resulting from maintaining full-precision embedding layers and storing quantization coefficients. Notably, **the methods using alternating update scheme offer a nearly $2 \times$ speedup over the Joint method**. This efficiency gain occurs because the Joint baseline doubles the equivalent batch size by simultaneously processing both forget and retain data in each step.
>
> ---
> **Q1. How sensitive is DO+ to the relative weighting between forgetting and retaining objectives?**
>
> **Reply:**  Since DualOptim+ adopts **alternating** update scheme and is based on Adam optimizer, **relative weighting between forgetting and retaining objectives does not affect the final performance**. To achieve a similar effect, **we can adjust the forget/retain frequency** ($F_f$ and $F_r$) instead. As reported in **Table 12** and **the tables in the response to Q1 of Reviewer gPLB**, DO+ is stable across different $F_r$ and achieves the best performance when $F_r=5$.
>
> ---
> **Q2. Could the authors comment on the memory overhead introduced by the additional optimizer states?**
>
> Please see the response to W3.
>
> ---
> **Q3. Do the authors expect the approach to provide similar benefits in other multi-objective optimization scenarios beyond unlearning?**
>
> Please see the response to W2.
>
> ---
> **Limitation**
>
> **Reply:** The limitation of this work is that we did not validate our method in more general tasks and we leave it as future work. We will indicate this in the revised version.
>
> ---
> **Reference**
>
> [1] Gradient Surgery for Multi-Task Learning
>
> [2] Conflict-Averse Gradient Descent for Multi-task Learning
>
> [3] Multi-Task Learning as a Bargaining Game
>
> [4] TRACE: A Comprehensive Benchmark for Continual Learning in Large Language Models

---

> > ### Author Rebuttal · Reviewer_fXAf · 2026-04-02
> >
> > I would like to thank the authors for their detailed rebuttal. After considering their response, I am maintaining my overall assessment as Weak Accept.

---

### Decision · Program_Chairs · 2026-04-30

**Decision:**

Accept (regular)

**Comment:**

This paper presents a technically sound and well-executed contribution to LLM unlearning. The key idea, bridging shared and decoupled optimizer states, is clean, intuitive, and practically useful. However, reviewers do raise concerns that the evaluation, while solid, does not fully substantiate broader claims of generality. That said, I am positive about this work and believe it provides a useful optimization perspective that could be impactful beyond the specific unlearning setting.